# The inexact power augmented Lagrangian method for constrained nonconvex optimization

**Alexander Bodard**                                              *alexander.bodard@kuleuven.be*
*ESAT-STADIUS & Leuven.AI, KU Leuven*

**Konstantinos Oikonomidis**                          *konstantinos.oikonomidis@kuleuven.be*
*ESAT-STADIUS & Leuven.AI, KU Leuven*

**Emanuel Laude**                                                    *elaude@proximafusion.com*
*Proxima Fusion GmbH*

**Panagiotis Patrinos**                                          *panos.patrinos@kuleuven.be*
*ESAT-STADIUS & Leuven.AI, KU Leuven*

**Reviewed on OpenReview:** *https: // openreview. net/ forum? id= 63ANb4r7EM*

## Abstract

This work introduces an unconventional inexact augmented Lagrangian method where the augmenting term is a Euclidean norm raised to a power between one and two. The proposed algorithm is applicable to a broad class of constrained nonconvex minimization problems that involve nonlinear equality constraints. In a first part of this work, we conduct a full complexity analysis of the method under a mild regularity condition, leveraging an accelerated first-order algorithm for solving the Hölder-smooth subproblems. Interestingly, this worst-case result indicates that using lower powers for the augmenting term leads to faster constraint satisfaction, albeit with a slower decrease of the dual residual. Notably, our analysis does not assume boundedness of the iterates. Thereafter, we present an inexact proximal point method for solving the weakly-convex and Hölder-smooth subproblems, and demonstrate that the combined scheme attains an improved rate that reduces to the best-known convergence rate whenever the augmenting term is a classical squared Euclidean norm. Different augmenting terms, involving a lower power, further improve the primal complexity at the cost of the dual complexity. Finally, numerical experiments validate the practical performance of unconventional augmenting terms.

## 1 Introduction

We consider *nonconvex* minimization problems with possibly *nonlinear* equality constraints of the form

$$\min_{x \in \mathbb{R}^n} \varphi(x) := f(x) + g(x) \quad \text{s.t.} \quad A(x) = 0. \tag{P}$$

Here $f : \mathbb{R}^n \to \mathbb{R}$ denotes a nonconvex continuously differentiable function, $g : \mathbb{R}^n \to \overline{\mathbb{R}}$ a proper, lsc and convex function, and $A : \mathbb{R}^n \to \mathbb{R}^m$ a continuously differentiable mapping. *Inequality* constraints $B(x) \leq 0$ can be incorporated into (P) by introducing slack variables $s \in \mathbb{R}^m$ subject to $B(x) + s = 0$ and $s \geq 0$.

The general formulation (P) incorporates a wide variety of problems arising in areas such as machine learning (Kulis et al., 2007; Ge et al., 2016; Tepper et al., 2018) and computer science (Ibaraki & Katoh, 1988; Zhao et al., 1998). Typical examples include generalized eigenvalue problems, nonconvex Burer-Monteiro reformulations (Burer & Monteiro, 2003; 2005), and Neyman-Pearson classification (Neyman et al., 1997), while recently this formulation has also been considered for neural network training (Sangalli et al., 2021; Evens et al., 2021; Liu et al., 2023).

This work focuses on *first-order methods* for solving (P), as they scale well with the problem dimensions and require significantly less memory compared to higher-order methods. These advantageous properties become increasingly important, particularly in applications that involve large amounts of data. Whereas existing works usually focus on problems with simple ('proximable') constraints, recent works also analyze first-order methods that handle more complicated constraints, see, e.g., (Xu, 2021; Li et al., 2021; Lin et al., 2022).

This paper proposes an unconventional *augmented Lagrangian method* (ALM) to solve (P) and analyzes its complexity under a mild regularity condition (cf. Assumption 4) while taking explicitly into account the inexact solutions of the subproblems. Notably, our framework is the first to conduct such an analysis in the presence of a generic convex term $g$ under Assumption 4 (cf. subsection 2.1), without assuming compactness of $\operatorname{dom}\varphi$ (cf. Lemma 2), and under relaxed smoothness assumptions.

## 1.1 Background and motivation

**Penalty and augmented Lagrangian methods** Penalty methods address (P) by solving, for some penalty function $\phi : \mathbb{R}^m \to \mathbb{R}$, a sequence of problems of the form

$$\underset{x \in \mathbb{R}^n}{\text{minimize}}\, \varphi(x) + \beta\phi(A(x)),$$

in which the penalty parameter $\beta > 0$ is gradually increased. A common penalty function is the squared Euclidean norm, i.e., $\phi(\cdot) = \frac{1}{2}\|\cdot\|^2$. However, it is well-known that quadratic penalty methods are empirically outperformed by ALMs. This last class of methods is based on the augmented Lagrangian (AL) function $L_\beta : \mathbb{R}^n \times \mathbb{R}^m \to \overline{\mathbb{R}}$, which, for a penalty parameter $\beta > 0$ and squared Euclidean norm $\phi$, reads

$$L_\beta(x, y) := \varphi(x) + \langle y, A(x)\rangle + \beta\phi(A(x)). \tag{1}$$

The AL involves a penalty term and an additional term $\langle y, A(x)\rangle$, where $y \in \mathbb{R}^m$ are called the dual variables or multipliers. Every classical ALM iteration updates the primal-dual pair $(x^k, y^k)$ by alternatingly minimizing $L_\beta$ with respect to the primal variable, and consecutively taking a dual ascent step on $L_\beta$, i.e.,

$$x^{k+1} \in \underset{x \in \mathbb{R}^n}{\arg\min}\, L_\beta(x, y^k), \quad y^{k+1} = y^k + \sigma A(x^{k+1}) \tag{2}$$

for some dual step size $\sigma \geq 0$. The primal ALM update can in general only be computed by means of an inner solver, and hence inexactly. In recent years, various works have taken this inexactness explicitly into account when analyzing the complexity of the method, which is then referred to as *inexact* ALM (iALM), see e.g., (Sahin et al., 2019).

**An unconventional penalty term** In this work we consider an iALM based on a generalization of the classical AL (1), in which the penalty term equals

$$\phi = \tfrac{1}{\nu+1}\|\cdot\|^{\nu+1}, \qquad \nu \in (0, 1]. \tag{3}$$

To distinguish from the classical setup $\nu = 1$, we refer to this as the *power* AL, and corresponding method.

**Connection to high-order proximal point methods** In the convex setting (Rockafellar, 1976) showed that classical ALM is equivalent to the *proximal point method* (PPM) applied to the negative Lagrange dual function $\varrho$, in the sense that $\{y^k\}_{k\in\mathbb{N}}$ generated by

$$y^{k+1} = \mathbf{prox}_{\beta\varrho}(y^k) := \underset{y \in \mathbb{R}^m}{\arg\min}\, \varrho(y) + \tfrac{1}{2\beta}\|y - y^k\|^2 \tag{4}$$

coincides with the sequence of multipliers defined by (2) with dual step size $\sigma = \beta$. More recently, (Oikonomidis et al., 2024b) showed that in the convex regime a similar dual interpretation exists for the so-called *power* ALM (2) (with $\phi$ as in (3)) if the dual update is modified to $y^{k+1} = y^k + \beta\nabla\phi(A(x^{k+1}))$, where

$$\nabla\phi(x) = \tfrac{1}{\|x\|^{1-\nu}}x \tag{5}$$

for all $x \neq 0$ and $\nabla \phi(0) = 0$. More precisely, the convex power ALM corresponds to a *high-order* PPM which is obtained by replacing the quadratic penalty in (4) with a higher power $\frac{1}{\beta^p(p+1)} \| \cdot \|_2^{p+1}$, $p \geq 1$ of the Euclidean norm $\| \cdot \|_2$ such that $\frac{1}{\nu+1} + \frac{1}{p+1} = 1$. The proposed nonconvex power augmented Lagrangian (1) reduces to the convex one when $\varphi$ is convex and $A$ is affine (Oikonomidis et al., 2024b, Example 2.1). Although the connection to higher-order methods does not extend to the nonconvex setup considered in this work, it motivates considering augmenting terms $\phi$ as in (3). Moreover, the rapid advancements regarding first-order methods for Hölder-smooth objectives over the past decade (Nesterov, 2015; Lan, 2015) potentially allow for faster solutions of the inner problems, which are Hölder smooth in this setting (cf. Lemma 3).

**Connection to sharp Lagrangians**    The proposed power AL interpolates between a classical AL with $\nu = 1$ and a *sharp* AL with $\nu = 0$ (Rockafellar & Wets, 1998, Example 11.58). The latter are appealing because they support exact penalty representations under mild conditions (Rockafellar & Wets, 1998, Example 11.62).

## 1.2   Related work

The augmented Lagrangian method, initially introduced in 1969 by Hestenes (1969); Powell (1969), is a popular algorithm that allows one to cast the constrained problem (P) into a sequence of smooth problems. Over the years it has been studied extensively, see e.g., the monographs (Bertsekas, 1982; Birgin & Martínez, 2014). ALM is also closely related to the popular Alternating Direction Method of Multipliers (Gabay & Mercier, 1976; Glowinski & Le Tallec, 1989), which has recently been studied for nonconvex equality constrained problems (Cohen et al., 2022; El Bourkhissi et al., 2023). In the convex regime, ALM was shown to be equivalent to the proximal point method (PPM) applied to the Lagrangian dual problem (Rockafellar, 1976). However, in the nonconvex regime this interpretation is lost at least globally (Rockafellar, 2023). Instead, we assume validity of a mild regularity condition (El Bourkhissi et al., 2025) (Assumption 4) to establish global convergence of the proposed scheme. An alternative approach that requires a (close to) feasible initial point, is explored by Grapiglia & Yuan (2021); Grapiglia (2023).

Inexact ALMs explicitly take into account inexactness in the primal update. They have been analyzed following the paper (Sahin et al., 2019), which establishes an $\widetilde{O}(\varepsilon^{-4})$ complexity for finding an $\varepsilon$-stationary point of (P) under a slightly different regularity condition – see subsection 2.1 for details. More recently, Li et al. (2021) employ a triple-loop scheme, based on an inexact proximal point method to solve the iALM subproblems, and obtain an improved complexity $\widetilde{O}(\varepsilon^{-3})$ under this regularity condition. This is the best-known rate of convergence of a first-order method for solving (P). Moreover, Lu (2022) proposes a single-loop primal-dual method that attains the same $\widetilde{O}(\varepsilon^{-3})$ complexity under a similar condition. We also mention that Lin et al. (2022) present a penalty method that attains the same $\widetilde{O}(\varepsilon^{-3})$ complexity.

To the best of our knowledge, the 'nonlinear' or higher-order proximal point method (using prox-term $\frac{1}{p+1} \| \cdot \|^{p+1}$, $p \geq 1$) was first studied in (Luque, 1987), which also analyzed its dual counterpart, the 'nonlinear' ALM (Luque, 1986). Besides being restricted to the convex case, these works provide only a local complexity analysis of the outer loop, neglecting the inherent difficulty of the Hölder-smooth subproblems – a challenge addressed by this paper. Recently, (Nesterov, 2023) studied the joint global complexity of high-order proximal point methods in the convex case; see also (Ahookhosh & Nesterov, 2021; 2024). Its dual counterpart, the power ALM for convex optimization has been recently studied in (Oikonomidis et al., 2024b).

Since the power augmented Lagrangian is the sum of a Hölder-smooth function (cf. Lemma 3) and a convex function, a first-order method that tackles such problems is desired. Besides the universal optimal methods by Nesterov (2015); Lan (2015) for convex problems, we highlight the accelerated methods by Ghadimi & Lan (2016); Ghadimi et al. (2019) for the convex and nonconvex setting respectively. Interesting developments in this area include the linesearch-free adaptive method by Li & Lan (2024) for convex problems, as well as a recent family adaptive proximal-gradient methods by Malitsky & Mishchenko (2020); Latafat et al. (2024) which were recently shown to also converge for locally Hölder-smooth problems (Oikonomidis et al., 2024a).

### 1.3 Contributions

We propose a novel inexact augmented Lagrangian method (iALM) with unconventional powers $\nu \in (0, 1]$ of the augmenting term $\phi$ (cf. (3)) for solving a general class of nonconvex problems with nonlinear constraints. The case $\nu = 1$ reduces to the standard iALM (Sahin et al., 2019). The complexity of the proposed method is analyzed, taking explicitly into account distinct primal and dual tolerances $\varepsilon_A, \varepsilon_\varphi > 0$. Notably,

- Under a mild regularity condition (Assumption 4) we prove convergence to first-order stationary points at a rate of $\omega^{-k}$ and show that the constraint violation decreases at a faster rate of $\omega^{-k/\nu}$, where $k$ is the number of (outer) iterations and $\omega$ determines the rate of increase of the penalty parameters. The joint complexity is then analyzed with the accelerated first-order method from (Ghadimi et al., 2019) as an inner solver for the nonconvex and Hölder-smooth subproblems. For $\nu = 1$, we subsume the complexity of (Sahin et al., 2019); for $\nu < 1$, we obtain *faster constraint satisfaction at the cost of a slower decrease in suboptimality*. Thus, the sharper penalties of the AL are reflected in the complexity analysis.

- Our complexity analysis improves upon existing results even for $\nu = 1$ by relaxing the standard Lipschitz smoothness assumption on $f$ to *local Hölder smoothness*, by *not assuming boundedness of the iterates*, and by including a *generic convex term $g$* in the objective.

- Under slightly more restrictive assumptions on $f$ and $A$, we present a *novel inexact proximal point method* to exploit the structure of the weakly-convex and Hölder-smooth subproblems. Using this inner solver, we further strengthen our complexity analysis to match the best-known $\widetilde{O}(\varepsilon^{-3})$ rate of convergence for the case $\nu = 1$. As before, $\nu < 1$ yields a better primal complexity at the cost of a worse dual complexity. The complexity is further improved when $A(x)$ is a linear mapping.

Finally, numerical simulations indicate that unconventional powers $\nu < 1$ also perform well in practice.

### 1.4 Notation, definitions, and technical assumptions

We denote the Euclidean inner product and norm on $\mathbb{R}^n$ by $\langle \cdot, \cdot \rangle$ and $\| \cdot \|$, respectively. For matrices $\| \cdot \|$ denotes the spectral norm. The Euclidean distance from a point $x$ to a set $\mathcal{X}$ is denoted by $\mathbf{dist}(x, \mathcal{X}) = \min_{z \in \mathcal{X}} \|x - z\|$. Given a differentiable mapping $A : \mathbb{R}^n \to \mathbb{R}^m$, we denote its Jacobian at $x$ by $J_A(x) \in \mathbb{R}^{m \times n}$. The $\widetilde{O}$-notation suppresses logarithmic dependencies. We denote by $\overline{\mathbb{R}} := \mathbb{R} \cup \{+\infty\}$ the extended real line. For an extended-valued function $g : \mathbb{R}^n \to \overline{\mathbb{R}}$ we denote by $\operatorname{dom} g = \{x \in \mathbb{R}^n \mid g(x) < \infty\}$ its domain and say that $g$ is proper if $\operatorname{dom} g \neq \emptyset$. A function is lower semi-continuous or simply lsc if its *epigraph* is closed (Rockafellar & Wets, 1998, §1B). For a convex function $g : \mathbb{R}^n \to \overline{\mathbb{R}}$ we denote by $\partial g(x)$ its (convex) subdifferential at $x \in \mathbb{R}^n$. A function $\varphi : \mathbb{R}^n \to \overline{\mathbb{R}}$ is called level-coercive if $\varphi$ is bounded below on bounded sets and satisfies $\liminf_{\|x\| \to \infty} \varphi(x)/\|x\| > 0$. For a closed, convex set $\mathcal{X}$ we define for any $x \in \mathbb{R}^n$ the normal cone of $\mathcal{X}$ at $x$ as $N_{\mathcal{X}}(x) = \{v \in \mathbb{R}^n \mid \langle v, y - x \rangle \leq 0, \forall y \in \mathcal{X}\}$, if $x \in \mathcal{X}$, and $N_{\mathcal{X}}(x) = \emptyset$ otherwise. We define by $\mathcal{C}^1(\mathbb{R}^n)$ the class of continuously differentiable functions on $\mathbb{R}^n$. We say that a mapping $F : \mathbb{R}^n \to \mathbb{R}^m$ is $H$-Hölder continuous of order $\nu \in (0, 1]$ on $\mathcal{X} \subseteq \mathbb{R}^n$ if $\|F(x') - F(x)\| \leq H\|x' - x\|^\nu$ for all $x, x' \in \mathcal{X}$ where $H > 0$. We say that a differentiable function $f : \mathbb{R}^n \to \mathbb{R}$ is $(H, \nu)$-Hölder smooth on $\mathcal{X}$ if $\nabla f(x)$ is $H$-Hölder-continuous of order $\nu$ on $\mathcal{X}$. When omitted, we assume $\mathcal{X} = \mathbb{R}^n$, and sometimes we simply write $\nu$-Hölder smooth. Finally, the function $f$ is $L_f$-Lipschitz smooth on $\mathcal{X}$ if it is $(L_f, 1)$-Hölder smooth on $\mathcal{X}$.

**Optimality conditions** If (P) has a local minimizer $x \in \mathbb{R}^n$ satisfying the constraint qualification

$$-J_A^\top(x)y \in N_{\operatorname{dom} g}(x), \qquad A(x) = 0 \qquad \implies \qquad y = 0, \tag{CQ}$$

then – this follows by applying the subdifferential chain rule (Rockafellar & Wets, 1998, Theorem 10.6) in a similar way as in Rockafellar & Wets (1998, Exercise 10.7) – there exists a vector $y \in \mathbb{R}^m$ such that

$$A(x) = 0, \qquad -\nabla f(x) - J_A^\top(x)y \in \partial g(x).$$

This set of (generalized) equations, describing the stationary points of (P), is naturally extended to accommodate approximately stationary points. In this paper we introduce two distinct tolerances for

**dist** $\left(-\nabla f(x) - J_A^\top(x)y, \partial g(x)\right)$ and $\|A(x)\|$, leading to the definition of $(\varepsilon_\varphi, \varepsilon_A)$-stationary points. We refer to $\varepsilon_A$ as the primal residual tolerance and to $\varepsilon_\varphi$ as the dual residual tolerance.

**Definition 1** (($\varepsilon_\varphi, \varepsilon_A$)-stationary points). *Given $\varepsilon_\varphi \geq 0$ and $\varepsilon_A \geq 0$, a point $x \in \mathbb{R}^n$ is called an $(\varepsilon_\varphi, \varepsilon_A)$-stationary point of* (P) *if there is a vector $y \in \mathbb{R}^m$ such that*

$$\|A(x)\| \leq \varepsilon_A, \qquad and \qquad \textbf{dist}\left(-\nabla f(x) - J_A^\top(x)y, \partial g(x)\right) \leq \varepsilon_\varphi. \tag{6}$$

Existing analyses for iALMs consider the case $\varepsilon_\varphi = \varepsilon_A > 0$, as the obtained complexities depend only on $\min\{\varepsilon_\varphi, \varepsilon_A\}$. In contrast, our analysis reflects that 'sharper' penalties with $\nu < 1$ yield faster constraint satisfaction (cf. Theorem 2), which nicely agrees with the exact penalty representation of sharp ALs (Rockafellar & Wets, 1998, Example 11.62). Also in practice, distinct primal and dual tolerances $\varepsilon_A, \varepsilon_\varphi > 0$ are relevant. Indeed, absolute tolerances are sensitive to a rescaling of the objective and constraints, and distinct tolerances can be used to compensate this. In fact, general-purpose solvers typically support distinct primal and dual tolerances for this reason. Moreover, it is not uncommon to analyze convergence for distinct primal and dual tolerances, see e.g. (Hermans et al., 2022) for a proximal ALM for quadratic programming problems. Yet, it appears unique that the primal and dual tolerance have a distinct effect on the worst-case complexity.

**Assumptions** Throughout this work, we make the following assumptions on $f$, $g$ and $A$.

**Assumption 1.** *For any nonempty compact set $\mathcal{S} \subseteq \text{dom } g$, there exist positive constants $H_f, H_A, A_{\max}, J_{A\max}, \nabla f_{\max}$ such that the following statements hold:*

    (i) *There exists an $\nu_f \in (0,1]$ such that $\|\nabla f(x') - \nabla f(x)\| \leq H_f\|x'-x\|^{\nu_f}$ for all $x, x' \in \mathcal{S}$;*

    (ii) *There exists an $\nu_A \in (0,1]$ such that $\|J_A(x') - J_A(x)\| \leq H_A\|x'-x\|^{\nu_A}$ for all $x, x' \in \mathcal{S}$;*

    (iii) $\|\nabla f(x)\| \leq \nabla f_{\max}$, $\|A(x)\| \leq A_{\max}$ *and $\|J_A(x)\| \leq J_{A\max}$ for all $x \in \mathcal{S}$.*

**Assumption 2.** *The function $g$ is proper, lsc and convex, and $\text{dom } g$ has nonempty interior. For any nonempty compact set $\mathcal{S} \subseteq \text{dom } g$, there exists $G \geq 0$ such that $\|g(x') - g(x)\| \leq G\|x'-x\|$ for all $x, x' \in \mathcal{S}$.*

**Assumption 3.** *The function $\varphi \equiv f + g$ is level-coercive.*

Assumption 1 is very mild: the conditions on $f$ hold when $f$ is differentiable with *locally* Hölder continuous gradients, and likewise the conditions on $A$ hold when $A$ has *locally* Hölder continuous Jacobians. Slightly more restrictive smoothness assumptions, i.e., with $\nu_f = \nu_A = 1$, have been used in various related works, see e.g., Sahin et al. (2019); Li et al. (2021); Lu (2022). Also Assumption 2 is very mild: a proper, lsc and convex function $g$ is Lipschitz continuous on any nonempty compact subset of $\text{int dom } g$ (Rockafellar, 1970, Theorem 24.7). Therefore, any potential issue arises at relative boundary points of the effective domain, as exemplified by the function $x \mapsto -\sqrt{x}$. We highlight in particular that indicators of closed convex sets, and real-valued convex functions all satisfy Assumption 2. Finally, note that $\varphi$ is level-coercive if either $f$ or $g$ is level-coercive while the other is bounded below (Rockafellar & Wets, 1998, Exercise 3.29). Alternatively, it suffices that $\varphi$ is coercive, as is the case when $\text{dom } \varphi \equiv \text{dom } g$ is bounded.

Notably, and contrary to related works (Sahin et al., 2019; Li et al., 2021; Lu, 2022; El Bourkhissi et al., 2025), no compactness of $\text{dom } \varphi$ or boundedness of the iterates is assumed, which makes the smoothness assumptions on $f$ and $A$ truly local. Moreover, we incorporate a generic convex term $g$. Also Sahin et al. (2019); El Bourkhissi et al. (2025) have done this, but, respectively, under a regularity condition which may not be ideal (cf. subsection 2.1) or under additional technical assumptions which may be hard to verify in practice. When analyzing the complexity of power ALM with the proximal point method as inner solver in section 3, we restrict our assumptions to match those of Li et al. (2021); Lu (2022). Table 1 summarizes our assumptions and compares them against those in similar works.

## 2 The inexact power augmented Lagrangian method

In this section we present the *inexact power augmented Lagrangian method* (power ALM), of which the pseudocode is shown in Algorithm 1. This method generalizes the inexact augmented Lagrangian method

Table 1: Comparison of the assumptions in this work against those in existing works.

| | Penalty term | Smoothness of $f$ and $A$ | Nonsmooth term $g$ | Boundedness of iterates | Regularity condition (RC) | Remark |
|---|---|---|---|---|---|---|
| (Sahin et al., 2019) | $\nu = 1$ | Lipschitz | generic convex | assumed | $\pm$ Assumption 4 | RC not ideal for generic $g$, cf. subsection 2.1 |
| (Li et al., 2021; Lu, 2022) | $\nu = 1$ | Lipschitz | convex indicator | compact dom $\varphi$ | Assumption 4 | |
| (El Bourkhissi et al., 2025) | $\nu = 1$ | Lipschitz | generic nonconvex | assumed | Assumption 4 | + technical conditions |
| Ours (Alg. 1) | $\nu \in (0,1]$ | Hölder (local) | generic convex | level-coercivity | Assumption 4 | |
| Ours (Alg. 1 + 2) | $\nu \in (0,1]$ | Lipschitz | convex indicator | compact dom $g$ | Assumption 4 | |

that was proposed and analyzed by Sahin et al. (2019); Li et al. (2021) to settings where $\nu \neq 1$. To adequately exploit the composite structure of the augmented Lagrangian $L_\beta(x,y)$, we define

$$\psi_\beta(x,y) := f(x) + \langle A(x), y \rangle + \frac{\beta}{2}\|A(x)\|^{\nu+1} \tag{7}$$

and note that the augmented Lagrangian $L_\beta(x,y) = \psi_\beta(x,y) + g(x)$ is the sum of a smooth and a nonsmooth term. The ALM subproblem in step 2 entails *inexactly* minimizing the augmented Lagrangian, i.e.,

---

**Algorithm 1** Inexact power augmented Lagrangian method

---

**Require:** $x^1 \in \mathbb{R}^n$, $y^1 \in \mathbb{R}^m$, $\lambda > 0$, $\omega > 1$, $\sigma_1, \beta_1 > 0$, $\nu \in (0,1]$.
1: **for** $k = 1, 2, \ldots$ **do**
2:     Update the tolerance $\varepsilon_{k+1} = \lambda/\beta_k$ and obtain $x^{k+1} \in \mathcal{X}$ such that

$$\mathbf{dist}(-\nabla_x \psi_{\beta_k}(x^{k+1}, y^k), \partial g(x^{k+1})) \leq \varepsilon_{k+1}. \tag{8}$$

3:     Update the dual step size as $\sigma_{k+1} = \sigma_1 \min(1, \frac{\|A(x^1)\|^\nu \log^2(2)}{\|A(x^{k+1})\|^\nu (k+1)\log^2(k+2)})$.
4:     Update the multipliers $y_{k+1} = y_k + \sigma_{k+1}\nabla\phi(A(x^{k+1}))$.
5:     Update the penalty parameter $\beta_{k+1} = \omega\beta_k$.
6: **end for**

---

$$\underset{x \in \mathbb{R}^n}{\text{minimize}}\, L_{\beta_k}(x, y^k) \equiv \psi_{\beta_k}(x, y^k) + g(x), \tag{9}$$

until the condition (8) is satisfied. Observe also that the dual step size update rule (step 3) is constructed in a way that ensures boundedness of the multipliers. This is captured by the following lemma.

**Lemma 1.** *The sequence $\{y^k\}_{k\in\mathbb{N}}$ generated by Algorithm 1 is bounded, i.e., there exists a $y_{\max} \in \mathbb{R}$, such that $\|y^k\| \leq y_{\max}$ for all $k \geq 1$.*

The proof is given in Appendix A.1. We remark that the results in this work can be extended to incorporate unbounded sequences of multipliers, by following the proofs of Li et al. (2021). The next lemma establishes that the iterates of Algorithm 1 remain in a compact set as long as the inner solver in step 2 monotonically decreases its objective. Hereby, and contrary to existing works (Sahin et al., 2019; Li et al., 2021; El Bourkhissi et al., 2025), no compactness assumption is required on dom $g$. A proof is given in Appendix A.1.

**Lemma 2.** *Suppose that Assumption 3 holds, and let $\{x^k\}_{k\in\mathbb{N}}, \{y^k\}_{k\in\mathbb{N}}, \{\beta^k\}_{k\in\mathbb{N}}$ be generated by Algorithm 1. For any $\beta \in \mathbb{R}, y \in \mathbb{R}^m, \bar{x} \in$ dom $g$, the sublevel set $\mathcal{L}_{\beta,y}(\bar{x}) := \{x \in \mathbb{R}^n : L_\beta(x,y) \leq L_\beta(\bar{x},y)\}$ is nonempty and compact. If, additionally, $L_{\beta_k}(x^{k+1}, y^k) \leq L_{\beta_k}(x^k, y^k)$ for $k \geq 1$, then:*

(i) *the sublevel sets satisfy $\mathcal{L}_{\beta_{k+1},y^{k+1}}(x^{k+1}) \subseteq \mathcal{L}_{\beta_k,y^k}(x^k)$ for $k \geq 1$;*

(ii) *the iterates $\{x^k\}_{k\in\mathbb{N}}$ remain in the (compact) initial sublevel set, i.e., $x^{k+1} \in \mathcal{L}_{\beta_1,y^1}(x^1)$ for $k \geq 1$.*

## 2.1 Complexity analysis

We analyze the computational complexity of Algorithm 1 under a regularity condition involving the nonlinear mapping $A$ and the normal cone $N_{\text{dom}\,g}$, which also naturally arises in the constraint qualification (CQ).

**Assumption 4** (regularity). *For any nonempty compact set $\mathcal{S} \subseteq \operatorname{dom} g$, there exists an $R > 0$ such that*

$$\operatorname{\mathbf{dist}}(-J_A^\top(x)A(x), N_{\operatorname{dom} g}(x)) \geq R\|A(x)\|, \qquad \textit{for all } x \in \mathcal{S}. \tag{R}$$

Assumption 4 was used by El Bourkhissi et al. (2025) to analyze the complexity of an ALM. If $g$ has full domain, then Assumption 4 reduces to a Polyak-Lojasiewicz-inequality on the feasibility problem $\operatorname{minimize}_x \frac{1}{2}\|A(x)\|^2$ (Sahin et al., 2019), and is a consequence of the uniform regularity condition by Bolte et al. (2018, Definition 3) in the so-called *information zone*. On the other hand, if $g = \delta_{\mathcal{X}}$ is the indicator of a closed convex set $\mathcal{X}$, and if the constraint qualification (CQ) – which is itself a generalization of the Mangasarian-Fromovitz condition (Rockafellar, 1993) – holds at a point $\bar{x} \in \mathbb{R}^n$, then (R) holds for the singleton $\mathcal{S} = \{\bar{x}\}$. Assumption 4 moreover assumes the existence of a *uniform constant $R > 0$* such that (R) holds for any $\mathcal{S} \in \operatorname{dom} g$. Existing works on iALMs, such as Li et al. (2021); Lu (2022), typically only deal with the case $g \equiv \delta_{\mathcal{X}}$, and analyze the corresponding complexity under Assumption 4. Sahin et al. (2019) use a condition similar to (R) involving the subdifferential $\partial g(x)$ instead of $N_{\operatorname{dom} g}(x)$. If $g = \delta_{\mathcal{X}}$, then this is equivalent because $\partial g(x) = N_{\operatorname{dom} g}(x) = N_{\mathcal{X}}(x)$ for $x \in \mathcal{X}$. However, if $g$ has full domain and is strictly continuous, it has been argued by El Bourkhissi et al. (2025) that Assumption 4 should not involve $g$, which is the case if $N_{\operatorname{dom} g}$ is used, but not for $\partial g$. We refer to El Bourkhissi et al. (2025, §5) for an extensive discussion and comparison to other regularity conditions.

Various problems satisfy Assumption 4, as shown by Sahin et al. (2019) for $g = \delta_{\mathcal{X}}$, including clustering, basis pursuit and others. Moreover, Li et al. (2021) demonstrate that affine equality constrained problems with an additional polyhedral constraint set or a ball constraint set also satisfy this condition. Some interesting constraint functions $A$ do not satisfy this condition globally. For example, (Bolte et al., 2018, Example 5) show that for $g \equiv 0$ and for spherical constraints $A$, (R) only holds if $\mathcal{S}$ is bounded away from the origin.

Our first result describes the number of power ALM iterations to obtain an approximate stationary point.

**Theorem 1** (Outer complexity). *Let $\{x^k\}_{k\in\mathbb{N}}$ denote the sequence of iterates generated by Algorithm 1. If Assumptions 1 to 4 hold, and if there exists a nonempty compact set $\mathcal{S} \subseteq \operatorname{dom} g$ containing the iterates $\{x^k\}_{k\in\mathbb{N}}$, then $x^{k+1}$ is a $\left(\frac{Q_f}{\beta_1 \omega^{k-1}}, \frac{Q_A}{\beta_1^{\frac{1}{\nu}} \omega^{\frac{k-1}{\nu}}}\right)$-stationary point of (P) with*

$$Q_f := \lambda + J_{A\max}\sigma_1 \frac{\nabla f_{\max} + G + J_{A\max}y_{\max} + \varepsilon_1}{R}, \quad Q_A := \left(\frac{\nabla f_{\max} + G + J_{A\max}y_{\max} + \varepsilon_1}{R}\right)^{1/\nu}. \tag{10}$$

The proof is given in Appendix A.1. We highlight that there exists a nonempty compact set $\mathcal{S}$ containing the iterates $\{x^k\}_{k\in\mathbb{N}}$ when the inner solver in step 2 of Algorithm 1 monotonically decreases its objective (cf. Lemma 2). If $\operatorname{dom} g$ is compact, then this condition is always satisfied, regardless of the inner solver.

Theorem 1 states that if such a nonempty compact set $\mathcal{S}$ exists, then Algorithm 1 finds a first order stationary point of (P) at a rate of $\omega^{-k}$, where $\omega > 1$ determines the rate of increase of the penalty parameters. This is identical to the result of Sahin et al. (2019) for iALM. Remarkably, in the case of Algorithm 1, the constraint violation $\|A(x^k)\|$ decreases at a *faster rate* of $\omega^{-k/\nu}$. However, it is important to note that the described rates are only in terms of the number of iterations of power ALM, i.e., the number of calls to the inner solver in step 2. To obtain a full complexity analysis of the method, we must specify the inner solver and require an estimate of its computational cost. This is related to the Hölder smoothness of $x \mapsto \psi_\beta(x,y)$.

**Lemma 3** (Augmented Lagrangian smoothness). *Let Assumption 1 hold. Then, for any $y \in \mathbb{R}^m$, and on any nonempty compact set $\mathcal{S} \subseteq \mathbb{R}^n$, the function $\psi_\beta(\cdot, y)$ as in (7) is $(H_\beta, q)$-Hölder smooth for some $H_\beta \geq 0$ which depends on $\mathcal{S}$, with $q = \min\{\nu_f, \nu_A, \nu\} \in (0, 1]$. In particular, we have for all $x, x' \in \mathcal{S}$ that*

$$\|\nabla_x \psi_\beta(x,y) - \nabla_x \psi_\beta(x',y)\| \leq H_\beta \|x - x'\|^q,$$

*where the modulus of Hölder smoothness $H_\beta$ depends on $H_f, H_A, J_{A_{\max}}$, and thus on $\mathcal{S}$, and is given by*

$$H_\beta = \left[H_f + H_A\|y\| + \beta(2^{1-\nu}J_{A\max}^{1+\nu} + A_{\max}^\nu H_A)\right]\max\{1, D^{1-q}\} \quad \textit{with} \quad D := \sup_{x,x'\in\mathcal{S}}\|x - x'\|.$$

The proof is given in Appendix A.1. The ALM subproblems of the form (9) in step 2 thus involve a composite objective, i.e., the sum of a nonconvex and Hölder-smooth function and a convex function. Minimizing such

an objective typically results in a higher computational cost than its counterpart with $\psi_\beta$ Lipschitz-smooth (Grimmer, 2024). The *unified problem-parameter free accelerated gradient* (UPFAG) method of Ghadimi et al. (2019) appears one of the only accelerated first-order methods for which a worst-case complexity analysis has been derived under both nonconvexity and Hölder smoothness. We proceed by deriving the total complexity of power ALM when UPFAG is used in step 2. For ease of notation we write $L_k(x) := L_{\beta_k}(x, y^k)$. Remark that the value $L_k^\star := \min_{x \in \mathbb{R}^n} L_k(x) > -\infty$ is finite, and hence the inner problems (9) in Algorithm 1 step 2 are well-defined. Indeed, from Assumptions 1 and 2 and Lemma 2, $L_k$ is proper, lsc and level-bounded. Lower boundedness then follows from (Rockafellar & Wets, 1998, Theorem 1.9). Moreover, since the UPFAG method enforces a monotonic decrease on its objective, Lemma 2 ensures that all UPFAG iterates remain in the compact initial sublevel set $\mathcal{L}_{\beta_1, y^1}(x^1)$. In the remainder of this section, we therefore define the smoothness constants from Assumptions 1 and 2 with respect to this set, without further mention.

The following lemma upper bounds the number of UPFAG iterations to inexactly solve an inner problem.

**Lemma 4** (Inner complexity). *Suppose that Assumptions 1 to 3 hold. Then the total number of (inner) iterations performed by the UPFAG method, with a small enough step size and initial iterate $x^k$, to obtain an $\varepsilon_{k+1}$-stationary point to the power ALM inner problem (cfr. Algorithm 1 step 2) is bounded by*

$$O\left( H_{\beta_k}^{1/q} \left[ \frac{L_k(x^k) - L_k^\star}{\varepsilon_{k+1}^2} \right]^{\frac{1+q}{2q}} \right). \tag{11}$$

This result follows by the complexity result in (Ghadimi et al., 2019, Corollary 5). Since we require a different termination criterion than the original work, we provide an explicit proof in Appendix A.1 for completeness. We can now describe the complexity of power ALM in terms of UPFAG iterations.

**Theorem 2** (Total complexity). *Suppose that Assumptions 1 to 4 hold and the UPFAG method from Ghadimi et al. (2019) is used for solving Algorithm 1 step 2 in the setting of Lemma 4. Given $\varepsilon_\varphi > 0$ and $\varepsilon_A > 0$, Algorithm 1 finds an $(\varepsilon_\varphi, \varepsilon_A)$-stationary point of (P) after at most $T = \max\{T_\varphi, T_A\}$ UPFAG iterations, where for $q = \min\{\nu_f, \nu_A, \nu\}$,*

$$T_\varphi = \widetilde{O}\left( \varepsilon_\varphi^{-\frac{5+3q}{2q}} \right), \qquad T_A = \widetilde{O}\left( \varepsilon_A^{-\frac{\nu}{q} - \frac{3\nu(1+q)}{2q}} \right). \tag{12}$$

The proof is given in Appendix A.1. In the Lipschitz smooth setting, our analysis subsumes the one from Sahin et al. (2019): if we choose $\nu = 1$ and $\varepsilon_\varphi = \varepsilon_A = \varepsilon$, we obtain $T_\varphi = T_A = \widetilde{O}\left( \varepsilon^{-4} \right)$. In the same setting, by choosing $\nu < 1$ we get a worse convergence rate for the dual residual and a better one for the primal. This result is in line with the intuition behind choosing a sharper augmenting term for the AL function, which highly penalizes the constraint violation. Finally, we highlight that the average number of gradient evaluations per UPFAG iteration is bounded by a constant (Ghadimi et al., 2019). It follows that Theorem 2 also describes the number of first-order oracle calls for finding an $(\varepsilon_\varphi, \varepsilon_A)$-stationary point of (P).

## 3 An inexact proximal point inner solver with improved complexity

This section presents an inexact proximal point method for solving the inner problems of Algorithm 1. The proposed scheme is essentially a double-loop algorithm that uses an accelerated gradient method for computing the proximal point updates, inspired by Kong et al. (2019); Li et al. (2021) and adapted to the Hölder smooth setting of our paper. However, we emphasize that this extension is not straightforward, since a Hölder-smooth subproblem cannot be made strongly convex by adding a sufficiently large quadratic term, and strong convexity of the inner-most problem is essential in obtaining an improved overall complexity. Henceforth we restrict Assumptions 1 to 3 as follows.

**Assumption 5.** *The function $g = \delta_\mathcal{X}$ is the indicator of a non-empty, convex and compact set $\mathcal{X} \subseteq \mathbb{R}^n$ with diameter $D > 0$. There exist positive constants $L_f, L_A, A_{\max}, J_{A\max}, \nabla f_{\max}$ such that:*

(i) $\|\nabla f(x') - \nabla f(x)\| \le L_f \|x' - x\|$ *for all* $x, x' \in \mathcal{X}$;

(ii) $\|J_A(x') - J_A(x)\| \le L_A \|x' - x\|$ *for all* $x, x' \in \mathcal{X}$;

(iii) $\|\nabla f(x)\| \leq \nabla f_{\max}$, $\|A(x)\| \leq A_{\max}$ and $\|J_A(x)\| \leq J_{A\max}$ for all $x \in \mathcal{X}$.

Under Assumption 5 we have $\nu_f = \nu_A = 1$, and hence by Lemma 3 the power AL has Hölder-continuous gradients of order $q = \nu$ on $\mathcal{X}$. Moreover, in this setting the power AL function is *weakly-convex*:

**Lemma 5.** *Suppose that Assumptions 4 and 5 hold. Then, for any $y \in \mathbb{R}^m$ the power augmented Lagrangian $L_\beta(\cdot, y)$ is $\rho$-weakly convex on $\mathcal{X}$, with $\rho := L_f + L_A(\|y\| + \beta A_{\max}^\nu)$.*

The proof is given in Appendix A.2. Therefore, every subproblem (9) in Algorithm 1 step 2 is of the form

$$\min_{x \in \mathcal{X}} \psi(x) := L_\beta(x, y), \tag{13}$$

for some $y \in \mathbb{R}^m$, where $\psi$ is $(H_\beta, \nu)$-Hölder-smooth and $\rho$-weakly convex on the compact set $\mathcal{X}$. The weak convexity of $\psi$ motivates the use of an inexact proximal point method, described in Algorithm 2. Note that the inexact proximal point updates entail minimizing a strongly convex and Hölder smooth function over a compact set and thus we can utilize the Fast Gradient Method (FGM) from Devolder et al. (2014) to compute them. Our approach differentiates from standard analyses in that the objective function of (13) has qualitatively distinct lower and upper bounds, obtained from the weak-convexity and Hölder-smoothness, respectively. To the best of our knowledge, the forthcoming analysis of the proposed inexact proximal point method is the first to exploit this, and as such enables an improved total complexity of Algorithms 1 and 2.

---

**Algorithm 2** Inexact proximal point method for (13)

---

**Require:** $x_1 \in \mathbb{R}^n$, tolerance $\varepsilon > 0$.
1: **for** $k = 1, 2, \dots$ **do**
2:     Let $F(\cdot) := \psi(\cdot) + \rho\| \cdot -x^k\|^2$ and obtain $x^{k+1} \in \mathcal{X}$ such that

$$\mathbf{dist}(-\nabla F(x^{k+1}), N_\mathcal{X}(x^{k+1})) \leq \varepsilon/4.$$

3:     **If** $2\rho\|x^{k+1} - x^k\| \leq \frac{\varepsilon}{2}$ **then** return $x^{k+1}$.
4: **end for**

---

### 3.1 Complexity analysis of the inexact proximal point method

The computation of a proximal point update, defined in Algorithm 2 step 2, involves a problem of the form

$$\min_{x \in \mathcal{X}} F(x) \tag{14}$$

where $F : \mathbb{R}^n \to \mathbb{R}$ is $(H_F, \nu)$-Hölder-smooth with $H_F := H_\beta + 2\rho \max\{1, D^{1-q}\}$, and $\rho$-strongly convex. We denote the minimizer of $F$ over $\mathcal{X}$ by $x^\star = \arg\min_{x \in \mathcal{X}} F(x)$. The following theorem describes the number of FGM iterations needed to obtain a point satisfying the inequality in Algorithm 2 step 2, and is based on (Devolder et al., 2014, §6.2). Its proof also handles the different termination criterion that we require compared to (Devolder et al., 2014, §6.2), and for this reason becomes rather technical.

**Theorem 3.** *Let $F$ be as in Algorithm 2 step 2 and suppose that Assumption 5 hold. Then, we need at most $T$ FGM iterations to obtain a point $x^+ \in \mathcal{X}$ that satisfies $\mathbf{dist}(-\nabla F(x^+), N_\mathcal{X}(x^+)) \leq \varepsilon$, where*

$$T = \widetilde{O}\left( \frac{H_F^{\frac{2}{1+3\nu}}}{\rho^{\frac{\nu+1}{3\nu+1}}} \overline{H}^{\frac{1+\nu}{\nu} \frac{1-\nu}{1+3\nu}} \varepsilon^{-\frac{1+\nu}{\nu} \frac{1-\nu}{1+3\nu}} \right), \qquad with \qquad \overline{H} = H_F^{1-\nu}(2H_F(1 + H_F))^{\frac{\nu}{2}} + (2H_F(1 + H_F))^{1/2}.$$

The proof is given in Appendix A.2. The next theorem, adapted from (Li et al., 2021, Theorem 1), provides an upper bound on the number of iterations the inexact proximal point method presented in Algorithm 2 needs in order to terminate. Its proof is also found in Appendix A.2.

**Theorem 4.** *Suppose that Assumption 5 holds. Then, Algorithm 2 stops within $T$ iterations, where $T = \left\lceil \frac{32\rho}{\varepsilon^2}(\psi(x^1) - \psi^\star) + 1 \right\rceil$, and the output $x^T \in \mathbb{R}^n$ satisfies $\mathbf{dist}(-\nabla \psi(x^T), N_\mathcal{X}(x^T)) \leq \varepsilon$.*

### 3.2 Joint complexity analysis of Algorithms 1 and 2

Having described the complexity of Algorithm 2 we now move on to the total complexity of the joint scheme, which is the main result of this section. We highlight that this result improves upon Theorem 2. For $\nu = 1$ it subsumes the complexity result in (Li et al., 2021, Theorem 2), whereas for $\nu < 1$ a better primal complexity is obtained at the cost of a worse dual complexity. A proof is given in Appendix A.2.

**Theorem 5** (Total complexity). *Suppose that Assumptions 4 and 5 hold, and let $\{x^k\}_{k \in \mathbb{N}}$ denote the iterates of Algorithm 1. If Algorithm 2 is used to solve the subproblems in Algorithm 1 step 2, and if the inexact proximal point updates in Algorithm 2 are computed using FGM, then an $(\varepsilon_\varphi, \varepsilon_A)$-stationary point of* (P) *is obtained after at most $T = \max\{T_\varphi, T_A\}$ FGM iterations, where*

$$T_\varphi = \widetilde{O}\left(\varepsilon_\varphi^{-3-\frac{1-\nu}{1+3\nu}\left(1+\frac{2(1+\nu)}{\nu}\right)}\right), \qquad T_A = \widetilde{O}\left(\varepsilon_A^{-3\nu-\frac{1-\nu}{1+3\nu}(3\nu+2)}\right).$$

As described in Appendix C, an FGM iteration requires a single gradient evaluation of $f$ and two projections onto $\mathcal{X}$. Consequently, Theorem 5 also describes the number of first-order oracle calls needed to find an $(\varepsilon_\varphi, \varepsilon_A)$-stationary point of (P).

### 3.3 Improved complexity for linear constraints

Theorem 5 describes the joint complexity of the triple-loop version of power ALM: Algorithm 1 in which the primal update in step 2 is obtained through Algorithm 2, and where in turn the inexact proximal point updates are computed using the FGM. If the constraint mapping $A$ is *linear*, this worst-case complexity can be further improved. It is obtained by following the exact same steps as in the proof of Theorem 5, and by remarking that if $A$ is linear, then $L_A = 0$. By Lemma 5, we have $\rho = L_f + L_A(\|y\| + \beta A_{\max}^\nu)$, and hence $\rho = L_f = O(1)$ if $A$ is linear, instead of $\rho = O(\beta)$ for nonlinear constraints. The following result subsumes the $\widetilde{O}\left(\varepsilon^{-\frac{5}{2}}\right)$ complexity of (Li et al., 2021, Theorem 2) for $\nu = 1$. A proof is given in Appendix A.2.

**Theorem 6** (Total complexity). *Suppose that the conditions of Theorem 5 hold, and additionally assume that the mapping $A$ is* linear. *If Algorithm 2 is used to solve the subproblems in Algorithm 1 step 2, and if the inexact proximal point updates in Algorithm 2 are computed using FGM, then an $(\varepsilon_\varphi, \varepsilon_A)$-stationary point of* (P) *is obtained after at most $T = \max\{T_\varphi, T_A\}$ FGM iterations, where*

$$T_\varphi = \widetilde{O}\left(\varepsilon_\varphi^{-2-2\frac{1-\nu}{1+3\nu}\frac{1+\nu}{\nu}-\frac{2}{1+3\nu}}\right), \qquad T_A = \widetilde{O}\left(\varepsilon_A^{-2\nu-2\frac{1-\nu}{1+3\nu}(1+\nu)-\frac{2\nu}{1+3\nu}}\right).$$

## 4 Numerical results

In this section, we compare the practical performance of power ALM (Algorithm 1) for various choices of the parameter $\nu \in (0, 1]$ to illustrate the empirical behavior of the proposed method in practice. Recall that the choice $\nu = 1$ reduces to the iALM from Sahin et al. (2019), and therefore is always included as a baseline comparison. All experiments are run in Julia on an HP EliteBook with 16 cores and 32 GB memory, and the source code is publicly available.[1] Unless mentioned otherwise, the subproblems in Algorithm 1 step 2 are solved using the UPFAG method of Ghadimi et al. (2019). In our experience, this approach works well in practice, and does not usually underperform when compared to the presented triple-loop scheme based on Algorithm 2, despite the improved worst-case convergence rate of the latter approach. Some additional experiments are included in Appendix B.

---

[1]https://github.com/alexanderbodard/tmlr_nonconvex_power_alm

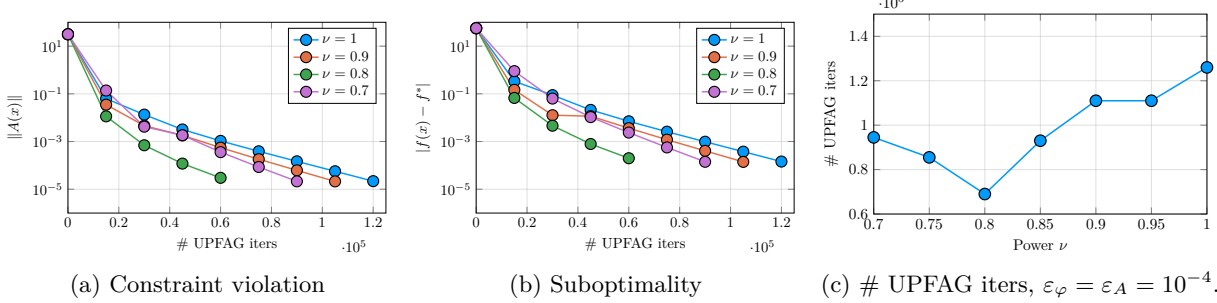

(a) Constraint violation     (b) Suboptimality     (c) # UPFAG iters, $\varepsilon_\varphi = \varepsilon_A = 10^{-4}$.

Figure 1: Comparison of power ALM with various powers $\nu$ on solving the clustering problem with Fashion-MNIST data Xiao et al. (2017). The case $\nu = 1$ corresponds to iALM from Sahin et al. (2019).

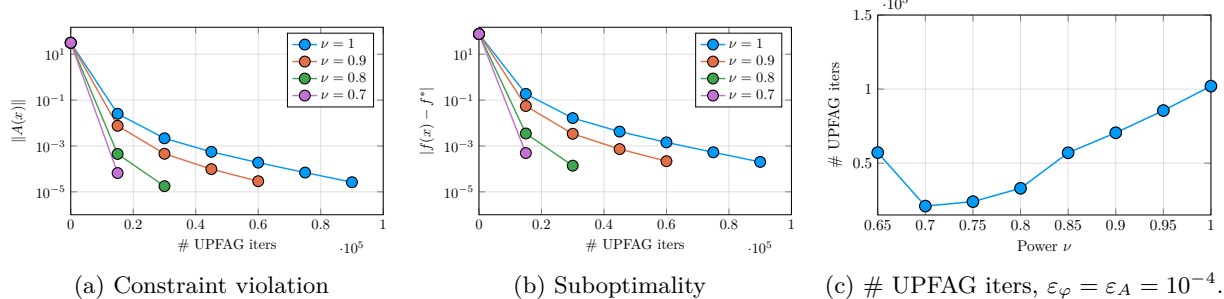

(a) Constraint violation     (b) Suboptimality     (c) # UPFAG iters, $\varepsilon_\varphi = \varepsilon_A = 10^{-4}$.

Figure 2: Comparison of power ALM with various powers $\nu$ on solving the clustering problem with MNIST data Deng (2012). The case $\nu = 1$ corresponds to iALM from Sahin et al. (2019).

## 4.1 Clustering

First, we consider a clustering problem which, following Sahin et al. (2019), can be reformulated in the form (P) through a rank-$r$ Burer-Monteiro relaxation with

$$f(x) = \sum_{i=1}^n \sum_{j=1}^n D_{i,j}\langle x_i, x_j \rangle, \quad A(x) = \left[ x_1^\top \sum_{i=1}^n x_i - 1, \ldots, x_n^\top \sum_{i=1}^n x_i - 1 \right]^\top.$$

Here $x := \left[ x_1^\top, \ldots x_n^\top \right]^\top \in \mathbb{R}^{rn}$ with $x_i \in \mathbb{R}^r$ for $i \in [1, n]$, and $\mathcal{X}$ is the intersection of the nonnegative orthant with the Euclidean ball of radius $\sqrt{s}$. The scalar $s$ denotes the number of clusters, and $D \in \mathbb{R}^{n \times n}$ is a distance matrix generated by some given data points $\{z^i\}_{i=1}^n$, i.e., such that $D_{i,j} = \|z^i - z^j\|$.

We test Algorithm 1 on two problem instances, being the MNIST dataset Deng (2012) and the Fashion-MNIST dataset Xiao et al. (2017). The setup is similar to that of Sahin et al. (2019), which is in turn based on Mixon et al. (2016). In particular, a simple two-layer neural network was used to first extract features from the data, and then this neural network was applied to $n = 1000$ random test samples from the dataset, yielding the vectors $\{z^i\}_{i=1}^{n=1000}$ that generate the distance matrix $D$. We define $s = 10, r = 20$, tune $\sigma_1 = 10$, $\lambda = 10^{-3}$, $\beta_1 = 5$, $\omega = 1.1$, and impose a maximum of $N = 1500$ UPFAG iterations per subproblem. Note that Sahin et al. (2019) showed that Assumption 4 is satisfied for this clustering problem. The results are visualized in Figures 1 and 2. We observe that the unconventional powers $\nu = 0.7$ and $\nu = 0.8$ perform best, requiring less than half the number of UPFAG iterations to converge to a $10^{-4}$-stationary point when compared to the iALM of Sahin et al. (2019) ($\nu = 1$).

## 4.2 Quadratic programs

Second, we consider nonconvex quadratic programs (QPs) of the form (P) with

$$f(x) = \frac{1}{2}x^\top Q x + \langle q, x \rangle, \quad \mathcal{X} = \{x \mid \underline{x} \le x \le \bar{x}\}, \quad A(x) = Cx - b.$$

Here $Q \in \mathbb{R}^{n \times n}$, $C \in \mathbb{R}^{m \times n}$, $b \in \mathbb{R}^m$, and $q \in \mathbb{R}^n$, with $Q$ symmetric and indefinite, are randomly generated, and $\underline{x}^i = -5$ and $\bar{x}^i = 5$ for $i \in \mathbb{N}_{[1,n]}$. We sample the entries of a diagonal matrix $\Lambda \in \mathbb{R}^{n \times n}$ from a Gaussian $\mathcal{N}(0, 50)$, and the entries of a matrix $\Sigma \in \mathbb{R}^{n \times n}$ from the standard Gaussian. Then, we normalize $\widehat{\Sigma} = \Sigma / \|\Sigma\|$ and define $Q := \widehat{\Sigma}^\top \Lambda \widehat{\Sigma}$. The entries of $q$ are sampled from a Gaussian $\mathcal{N}(0, 2)$, and the entries of $C$ are sampled from the standard Gaussian. We construct a vector $\mu \in \mathbb{R}^n$ with standard Gaussian entries, and define $b := C\mu$. A maximum of $N = 10^4$ UPFAG iterations per subproblem is imposed, and the parameters $\beta_1 = 5, \omega = 1.1, \lambda = 10^{-3}, \sigma_1 = 10$ follow the tuning from the previous subsection. We use tolerances $\varepsilon_\varphi = \varepsilon_A = 10^{-3}$. Remark that the AL function with penalty $\beta_k$ has $\|Q + \beta_k C^\top C\|$-Lipschitz continuous gradients for $\nu = 1$. We highlight that the dual residual can be efficiently computed by following the procedure described by (Nedelcu et al., 2014, Eq. 2.7 and below).

We generate 100 random QP instances of size $n = 200, m = 10$, and visualize the total number of UPFAG iterations and the objective value at the returned points by means of violin plots in Figure 3. We observe that $\nu = 0.8$ and $\nu = 0.9$ perform best in terms of UPFAG iterations. Yet, we also note the more robust performance of $\nu = 0.6$. Regarding the quality of the (local) solutions, Figure 3b indicates that the use of unconventional powers $\nu$ does not yield significantly better or worse solutions than the classical choice $\nu = 1$.

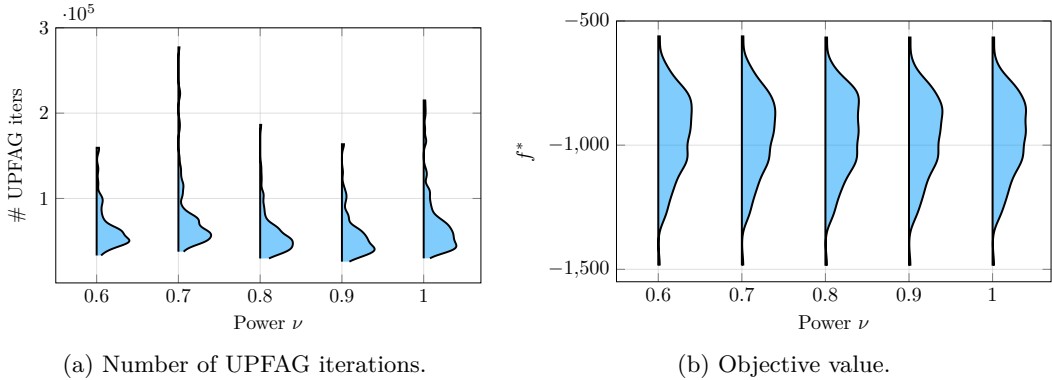

(a) Number of UPFAG iterations.

(b) Objective value.

Figure 3: Power ALM for various powers $\nu$ on solving 100 random QPs of size $n = 100, m = 20$.

Table 2 reports the number of UPFAG iterations performed by power ALM for the first 10 randomly generated QPs, and also includes the obtained primal and dual residuals. The power $\nu = 0.8$ performs well compared to the classical setup $\nu = 1$, and requires roughly 25% fewer inner iterations on average. Note that the best primal residuals are attained by the smaller powers $\nu$, and the best dual residuals by the higher powers $\nu$. For $\nu = 0.6$ and $\nu = 0.7$, the inner solver often attains the maximum number of inner iterations $N$. In our experience, further increasing $N$ for these powers works counterproductively and increases the total number of UPFAG iterations.

Table 2: Power ALM with UPFAG inner solver on solving random QPs of size $n = 100, m = 20$.

| | $\nu = 0.6$ | | | $\nu = 0.7$ | | | $\nu = 0.8$ | | | $\nu = 0.9$ | | | $\nu = 1.0$ | | |
|---|---|---|---|---|---|---|---|---|---|---|---|---|---|---|---|
| trial | pres | dres | # iters | pres | dres. | # iters | pres | dres | # iters | pres | dres | # iters | pres | dres | # iters |
| 1 | $3.7 \cdot 10^{-4}$ | $7.5 \cdot 10^{-7}$ | 75,063 | $5.8 \cdot 10^{-4}$ | $1.3 \cdot 10^{-6}$ | 84,362 | $9.8 \cdot 10^{-4}$ | $1.9 \cdot 10^{-6}$ | 64,649 | $5.6 \cdot 10^{-4}$ | $2.7 \cdot 10^{-6}$ | 48,409 | $7.2 \cdot 10^{-4}$ | $1 \cdot 10^{-6}$ | 58,903 |
| 2 | $1.9 \cdot 10^{-4}$ | $8 \cdot 10^{-7}$ | 75,740 | $6.9 \cdot 10^{-4}$ | $6.3 \cdot 10^{-6}$ | 80,000 | $9 \cdot 10^{-4}$ | $7.1 \cdot 10^{-7}$ | 60,237 | $5.4 \cdot 10^{-4}$ | $1.4 \cdot 10^{-6}$ | 55,371 | $6.9 \cdot 10^{-4}$ | $9.1 \cdot 10^{-7}$ | 68,320 |
| 3 | $9.9 \cdot 10^{-4}$ | $6.3 \cdot 10^{-7}$ | $1.3 \cdot 10^5$ | $8 \cdot 10^{-4}$ | $5.9 \cdot 10^{-7}$ | $1.8 \cdot 10^5$ | $9.4 \cdot 10^{-4}$ | $8.3 \cdot 10^{-7}$ | $1.2 \cdot 10^5$ | $8.3 \cdot 10^{-4}$ | $6 \cdot 10^{-7}$ | $1 \cdot 10^5$ | $8.7 \cdot 10^{-4}$ | $6.5 \cdot 10^{-7}$ | $1.3 \cdot 10^5$ |
| 4 | $7.5 \cdot 10^{-4}$ | $3.4 \cdot 10^{-6}$ | 50,000 | $8.4 \cdot 10^{-4}$ | $3 \cdot 10^{-6}$ | 60,000 | $9.4 \cdot 10^{-4}$ | $1.6 \cdot 10^{-6}$ | 69,593 | $9.3 \cdot 10^{-4}$ | $1.1 \cdot 10^{-6}$ | 66,592 | $8.8 \cdot 10^{-4}$ | $2.5 \cdot 10^{-6}$ | $1.1 \cdot 10^5$ |
| 5 | $8.6 \cdot 10^{-5}$ | $4.4 \cdot 10^{-5}$ | 50,000 | $6.4 \cdot 10^{-4}$ | $4.2 \cdot 10^{-5}$ | 50,000 | $1.1 \cdot 10^{-4}$ | $4.7 \cdot 10^{-6}$ | 42,728 | $8.6 \cdot 10^{-4}$ | $3.3 \cdot 10^{-6}$ | $1.6 \cdot 10^5$ | $9.7 \cdot 10^{-4}$ | $7 \cdot 10^{-7}$ | $2.1 \cdot 10^5$ |
| 6 | $2.8 \cdot 10^{-4}$ | $1.5 \cdot 10^{-6}$ | 55,738 | $9.7 \cdot 10^{-4}$ | $1.4 \cdot 10^{-6}$ | 57,214 | $4.1 \cdot 10^{-4}$ | $1.3 \cdot 10^{-6}$ | 51,055 | $6.9 \cdot 10^{-4}$ | $1.4 \cdot 10^{-6}$ | 45,508 | $8 \cdot 10^{-4}$ | $1.4 \cdot 10^{-6}$ | 51,917 |
| 7 | $9.5 \cdot 10^{-4}$ | $9.6 \cdot 10^{-5}$ | 40,000 | $4.7 \cdot 10^{-4}$ | $7.3 \cdot 10^{-6}$ | 50,000 | $2.3 \cdot 10^{-4}$ | $3.2 \cdot 10^{-5}$ | 41,242 | $9.4 \cdot 10^{-4}$ | $1.4 \cdot 10^{-6}$ | 37,795 | $5.9 \cdot 10^{-4}$ | $3.9 \cdot 10^{-6}$ | 42,767 |
| 8 | $5.5 \cdot 10^{-4}$ | $1.5 \cdot 10^{-5}$ | 60,000 | $5.7 \cdot 10^{-4}$ | $1.4 \cdot 10^{-5}$ | 70,000 | $7.6 \cdot 10^{-4}$ | $1.4 \cdot 10^{-6}$ | 68,184 | $5.8 \cdot 10^{-4}$ | $1.1 \cdot 10^{-6}$ | 74,075 | $6.8 \cdot 10^{-4}$ | $8.2 \cdot 10^{-7}$ | 87,036 |
| 9 | $4.7 \cdot 10^{-4}$ | $8.9 \cdot 10^{-5}$ | 50,000 | $3.3 \cdot 10^{-4}$ | $1.4 \cdot 10^{-6}$ | 60,000 | $3.8 \cdot 10^{-4}$ | $1.9 \cdot 10^{-6}$ | 48,242 | $4.5 \cdot 10^{-4}$ | $1.2 \cdot 10^{-6}$ | 42,656 | $9.8 \cdot 10^{-4}$ | $7.3 \cdot 10^{-6}$ | 44,826 |
| 10 | $5.6 \cdot 10^{-4}$ | $2.9 \cdot 10^{-6}$ | 61,594 | $6.7 \cdot 10^{-4}$ | $2 \cdot 10^{-6}$ | 72,538 | $5.1 \cdot 10^{-4}$ | $1.9 \cdot 10^{-6}$ | 54,563 | $5.3 \cdot 10^{-4}$ | $1.9 \cdot 10^{-6}$ | 46,028 | $6.7 \cdot 10^{-4}$ | $1.2 \cdot 10^{-6}$ | 55,466 |
| avg. | $\mathbf{5.2 \cdot 10^{-4}}$ | $2.5 \cdot 10^{-5}$ | 65,060.5 | $6.6 \cdot 10^{-4}$ | $8 \cdot 10^{-6}$ | 76,583.8 | $6.2 \cdot 10^{-4}$ | $4.8 \cdot 10^{-6}$ | $\mathbf{61,940.3}$ | $6.9 \cdot 10^{-4}$ | $\mathbf{1.6 \cdot 10^{-6}}$ | 68,343 | $7.9 \cdot 10^{-4}$ | $2 \cdot 10^{-6}$ | 85,558.3 |

We now compare these results to Table 3, which is similar to Table 2, but uses the triple loop version of power ALM, i.e., Algorithm 1 in which the primal update in step 2 is computed using Algorithm 2 (iPPM). The inexact proximal point updates are computed using the accelerated adaptive proximal-gradient

algorithm presented in (Malitsky & Mishchenko, 2020, Algorithm 2), which, although a heuristic, significantly outperforms the other methods we tried. We tuned $\beta_1 = 1, \omega = 1.1, \lambda = 0.01$. Although the triple-loop version of power ALM has a better worst-case complexity, in practice we find that it is usually outperformed by its double-loop variant, as confirmed in this table. We do, nevertheless, observe that the triple-loop method is more stable, in the sense that the number of gradient calls fluctuates less between QP realizations. This was also observed for $\nu = 1$ in Li et al. (2021). Moreover, remark that as $\nu$ decreases, the primal residual or constraint violation also decreases, thus qualitatively confirming our theoretical results that decreasing $\nu$ yields faster constraint satisfaction.

Table 3: Power ALM with iPPM inner solver on solving random QPs of size $n = 100, m = 20$.

| | $\nu = 0.6$ | | | $\nu = 0.7$ | | | $\nu = 0.8$ | | | $\nu = 0.9$ | | | $\nu = 1.0$ | | |
|---|---|---|---|---|---|---|---|---|---|---|---|---|---|---|---|
| trial | pres | dres | # grads | pres | dres. | # grads | pres | dres | # grads | pres | dres | # grads | pres | dres | # grads |
| 1 | $1.4 \cdot 10^{-5}$ | $9.7 \cdot 10^{-4}$ | $1.2 \cdot 10^{5}$ | $3.8 \cdot 10^{-5}$ | $8.4 \cdot 10^{-4}$ | $1.2 \cdot 10^{5}$ | $1.2 \cdot 10^{-5}$ | $9.4 \cdot 10^{-4}$ | $1.4 \cdot 10^{5}$ | $9.3 \cdot 10^{-6}$ | $1 \cdot 10^{-3}$ | $1.3 \cdot 10^{5}$ | $1.3 \cdot 10^{-4}$ | $9.4 \cdot 10^{-4}$ | $1.1 \cdot 10^{5}$ |
| 2 | $1.3 \cdot 10^{-5}$ | $9.4 \cdot 10^{-4}$ | $1.3 \cdot 10^{5}$ | $4.7 \cdot 10^{-4}$ | $7 \cdot 10^{-4}$ | $1.2 \cdot 10^{5}$ | $2.2 \cdot 10^{-4}$ | $8.1 \cdot 10^{-4}$ | $1.1 \cdot 10^{5}$ | $3.6 \cdot 10^{-4}$ | $5.2 \cdot 10^{-4}$ | $93{,}423$ | $8.8 \cdot 10^{-4}$ | $5.4 \cdot 10^{-4}$ | $1 \cdot 10^{5}$ |
| 3 | $8.7 \cdot 10^{-6}$ | $6.3 \cdot 10^{-4}$ | $1.3 \cdot 10^{5}$ | $2.8 \cdot 10^{-5}$ | $5.4 \cdot 10^{-4}$ | $1.4 \cdot 10^{5}$ | $2 \cdot 10^{-4}$ | $8.8 \cdot 10^{-4}$ | $1.4 \cdot 10^{5}$ | $6.1 \cdot 10^{-5}$ | $9.5 \cdot 10^{-4}$ | $1.6 \cdot 10^{5}$ | $3.8 \cdot 10^{-5}$ | $9.3 \cdot 10^{-4}$ | $1.8 \cdot 10^{5}$ |
| 4 | $1.6 \cdot 10^{-7}$ | $9.1 \cdot 10^{-4}$ | $2.9 \cdot 10^{6}$ | $5.7 \cdot 10^{-5}$ | $7.2 \cdot 10^{-4}$ | $1 \cdot 10^{5}$ | $1.3 \cdot 10^{-4}$ | $4.8 \cdot 10^{-4}$ | $99{,}070$ | $1.8 \cdot 10^{-4}$ | $7.9 \cdot 10^{-4}$ | $87{,}422$ | $7.1 \cdot 10^{-4}$ | $7.1 \cdot 10^{-4}$ | $1.2 \cdot 10^{5}$ |
| 5 | $9.5 \cdot 10^{-8}$ | $9.7 \cdot 10^{-4}$ | $1.7 \cdot 10^{6}$ | $2.9 \cdot 10^{-7}$ | $9.6 \cdot 10^{-4}$ | $1.9 \cdot 10^{5}$ | $1.5 \cdot 10^{-4}$ | $8.1 \cdot 10^{-4}$ | $2.2 \cdot 10^{5}$ | $1.7 \cdot 10^{-4}$ | $3 \cdot 10^{-4}$ | $1.6 \cdot 10^{5}$ | $5.4 \cdot 10^{-4}$ | $8.7 \cdot 10^{-4}$ | $1.9 \cdot 10^{5}$ |
| 6 | $5 \cdot 10^{-8}$ | $9.8 \cdot 10^{-4}$ | $9.5 \cdot 10^{5}$ | $8.6 \cdot 10^{-7}$ | $1 \cdot 10^{-3}$ | $1.6 \cdot 10^{5}$ | $4.1 \cdot 10^{-6}$ | $9.9 \cdot 10^{-4}$ | $1.3 \cdot 10^{5}$ | $1.8 \cdot 10^{-5}$ | $9.4 \cdot 10^{-4}$ | $1.2 \cdot 10^{5}$ | $7.6 \cdot 10^{-4}$ | $9.6 \cdot 10^{-4}$ | $1.3 \cdot 10^{5}$ |
| 7 | $1 \cdot 10^{-6}$ | $8.8 \cdot 10^{-4}$ | $1.3 \cdot 10^{6}$ | $6.6 \cdot 10^{-6}$ | $9.4 \cdot 10^{-4}$ | $2.5 \cdot 10^{5}$ | $2.9 \cdot 10^{-5}$ | $9.3 \cdot 10^{-4}$ | $1.3 \cdot 10^{5}$ | $1.1 \cdot 10^{-4}$ | $8.5 \cdot 10^{-4}$ | $1.6 \cdot 10^{5}$ | $2.7 \cdot 10^{-4}$ | $8.8 \cdot 10^{-4}$ | $1.6 \cdot 10^{5}$ |
| 8 | $2.1 \cdot 10^{-5}$ | $4.6 \cdot 10^{-4}$ | $95{,}543$ | $1.6 \cdot 10^{-4}$ | $5.9 \cdot 10^{-4}$ | $1.2 \cdot 10^{5}$ | $8.7 \cdot 10^{-4}$ | $2.5 \cdot 10^{-4}$ | $3.2 \cdot 10^{5}$ | $5.2 \cdot 10^{-4}$ | $7.1 \cdot 10^{-4}$ | $1.2 \cdot 10^{5}$ | $8.1 \cdot 10^{-5}$ | $8.8 \cdot 10^{-4}$ | $1.2 \cdot 10^{5}$ |
| 9 | $8.5 \cdot 10^{-7}$ | $9.7 \cdot 10^{-4}$ | $8.7 \cdot 10^{5}$ | $2.5 \cdot 10^{-6}$ | $9.8 \cdot 10^{-4}$ | $3 \cdot 10^{5}$ | $2.7 \cdot 10^{-5}$ | $9.6 \cdot 10^{-4}$ | $1.1 \cdot 10^{5}$ | $4 \cdot 10^{-5}$ | $8.3 \cdot 10^{-4}$ | $1.3 \cdot 10^{5}$ | $1.6 \cdot 10^{-4}$ | $8.9 \cdot 10^{-4}$ | $1.6 \cdot 10^{5}$ |
| 10 | $2.7 \cdot 10^{-4}$ | $4.2 \cdot 10^{-4}$ | $89{,}998$ | $6.3 \cdot 10^{-5}$ | $7 \cdot 10^{-4}$ | $87{,}063$ | $1.7 \cdot 10^{-5}$ | $9 \cdot 10^{-4}$ | $1.1 \cdot 10^{5}$ | $9.5 \cdot 10^{-5}$ | $8.4 \cdot 10^{-4}$ | $1.6 \cdot 10^{5}$ | $7.8 \cdot 10^{-5}$ | $9.1 \cdot 10^{-4}$ | $1.1 \cdot 10^{5}$ |
| avg. | $\mathbf{3.3 \cdot 10^{-5}}$ | $8.1 \cdot 10^{-4}$ | $8.4 \cdot 10^{5}$ | $8.3 \cdot 10^{-5}$ | $8 \cdot 10^{-4}$ | $1.6 \cdot 10^{5}$ | $1.7 \cdot 10^{-4}$ | $7.9 \cdot 10^{-4}$ | $1.5 \cdot 10^{5}$ | $1.6 \cdot 10^{-4}$ | $\mathbf{7.7 \cdot 10^{-4}}$ | $\mathbf{1.3 \cdot 10^{5}}$ | $3.6 \cdot 10^{-4}$ | $8.5 \cdot 10^{-4}$ | $1.4 \cdot 10^{5}$ |

## 5 Conclusion

In this paper we introduced and analyzed an inexact augmented Lagrangian method for nonconvex problems with nonlinear constraints, involving a potentially sharper penalty function. Taking into account the composite structure of the corresponding augmented Lagrangian function, we study the joint complexity of the scheme using two different subproblem oracles. One of these oracles is a novel proximal point scheme that exploits the specific structure of the subproblems, of which the objectives are Hölder smooth and weakly convex. The proposed augmented Lagrangian method generalizes existing works with conventional penalty terms that attain the best known convergence rate for nonconvex minimization with first-order methods. Notably, we proved that unconventional penalty terms yield faster constraint satisfaction at the cost of a slower decrease of the cost, thereby reflecting the sharper penalties in the complexity analysis. It is noteworthy that our analysis also improves upon existing works by considering a generic convex term $g$ in the objective, and by not assuming boundedness of the iterates. Numerical experiments indicate that also in practice such penalty terms perform well.

### Acknowledgments

This work was supported by: Research Foundation Flanders (FWO) research projects G081222N, G033822N, G0A0920N; Research Council KU Leuven C1 project No. C14/24/103; and EuroHPC Project: 101118139 Inno4Scale.

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

## A Proofs

### A.1 Proofs of section 2

**Proof of Lemma 1**

*Proof.* By consecutively using the multiplier update (step 4), the triangle inequality, and the dual step size update rule (step 3) we obtain that

$$\|y^{k+1}\| = \left\| y^1 + \sum_{i=2}^{k} \sigma_i A(x^i) \|A(x^i)\|^{q-1} \right\|$$

$$\leq \|y^1\| + \sum_{i=2}^{k} \sigma_i \|A(x^i)\|^q$$

$$\leq \|y^1\| + \sum_{i=2}^{k} \sigma_1 \frac{\|A(x^1)\|(\log 2)^2}{i\left(\log(i+1)\right)^2}$$

$$= \|y^1\| + c\sigma_1 \|A(x^1)\|(\log 2)^2 =: y_{\max},$$

where $c = \sum_{i=2}^{\infty} \frac{1}{i \log^2(i+1)} < \infty$. $\square$

**Proof of Lemma 2**

We first establish the following lemma.

**Lemma 6.** *Let $\{x^k\}_{k\in\mathbb{N}}, \{y^k\}_{k\in\mathbb{N}}, \{\beta^k\}_{k\in\mathbb{N}}$ be generated by Algorithm 1. Then for any $x \in \mathrm{dom}\, g$ and $k \geq 1$,*

$$\mathcal{L}_{\beta_{k+1},y^{k+1}}(x) \subseteq \mathcal{L}_{\beta_k,y^k}(x)$$

*Proof.* From the definition of the augmented Lagrangian, the multiplier update and the penalty parameter update with $\omega > 1$ we have for $k \geq 1$

$$L_{\beta_{k+1},y^{k+1}}(x) = \varphi(x) + \langle A(x), y^{k+1} \rangle + \frac{\beta_{k+1}}{2}\|A(x)\|^{\nu+1}$$

$$= \varphi(x) + \langle A(x), y^k \rangle + \sigma_{k+1}\|A(x)\|^{1+\nu} + \omega\frac{\beta_k}{2}\|A(x)\|^{\nu+1}$$

$$\geq \varphi(x) + \langle A(x), y^k \rangle + \frac{\beta_k}{2}\|A(x)\|^{\nu+1}$$

$$= L_{\beta_k,y^k}(x).$$

It follows immediately that

$$\mathcal{L}_{\beta_{k+1},y^{k+1}}(x) \subseteq \mathcal{L}_{\beta_k,y^k}(x).$$

$\square$

We now provide a proof for Lemma 2.

*Proof.* The nonemptyness claim follows from the observation that $\bar{x} \in \mathcal{L}_{\beta,y}(\bar{x})$ always holds by definition of the sublevel set. For the compactness claim, recall that the augmented Lagrangian can be expressed as $L_\beta(x,y) = \varphi(x) + \langle A(x), y \rangle + \frac{\beta}{2}\|A(x)\|^{\nu+1}$. By (Rockafellar & Wets, 1998, Corollary 3.27), it suffices to establish level-coercivity of $L_\beta(x,y)$ with respect to the primal variable. A sufficient condition for this is that $\varphi$ is level-coercive, as guaranteed by Assumption 3, and that $x \mapsto \langle A(x), y \rangle + \frac{\beta}{2}\|A(x)\|^\nu$ is bounded below (Rockafellar & Wets, 1998, Exercise 3.29 (b)). Note that $\langle A(x), y \rangle + \frac{\beta}{2}\|A(x)\|^{\nu+1} \geq -\|A(x)\|\|y\| + \frac{\beta}{2}\|A(x)\|^{\nu+1}$. If $\|A(x)\|^\nu \leq \frac{2}{\beta}\|y\|$, the claimed boundedness follows from the fact that

$$\langle A(x), y \rangle + \frac{\beta}{2}\|A(x)\|^{\nu+1} \geq -\left(\frac{2}{\beta}\|y\|\right)^{\frac{1}{\nu}}\|y\| + \frac{\beta}{2}\|A(x)\|^{\nu+1} \geq -\left(\frac{2}{\beta}\|y\|\right)^{\frac{1}{\nu}}\|y\|.$$

Else $\|A(x)\|^\nu > \frac{2}{\beta}\|y\|$, and hence

$$\langle A(x), y \rangle + \frac{\beta}{2}\|A(x)\|^{\nu+1} \geq -\|A(x)\|\|y\| + \frac{\beta}{2}\frac{2}{\beta}\|A(x)\|\|y\| = 0,$$

which also implies the claimed boundedness.

We now show Lemma 2(i) by induction. First, observe that the assumption $L_{\beta_k}(x^{k+1}, y^k) \leq L_{\beta_k}(x^k, y^k)$ implies for $k \geq 1$ that

$$\mathcal{L}_{\beta_k, y^k}(x^{k+1}) \subseteq \mathcal{L}_{\beta_k, y^k}(x^k). \tag{15}$$

The base case $k = 1$ follows by consecutively applying Lemma 6 and (15), i.e.,

$$\mathcal{L}_{\beta_2, y^2}(x^2) \subseteq \mathcal{L}_{\beta_1, y^1}(x^2) \subseteq \mathcal{L}_{\beta_1, y^1}(x^1)$$

Suppose that Lemma 2(i) holds for some $k \geq 1$. Then by Lemma 6 we have

$$\mathcal{L}_{\beta_{k+1}, y^{k+1}}(x^{k+1}) \subseteq \mathcal{L}_{\beta_k, y^k}(x^{k+1}) \subseteq \mathcal{L}_{\beta_k, y^k}(x^k),$$

where again we applied Lemma 6 and (15). This establishes Lemma 2(i).

Finally, Lemma 2(ii) follows by consecutive application of Lemma 2(i) and by compactness of $\mathcal{L}_{\beta_1, y^1}(x^1)$. $\quad\square$

**Proof of Theorem 1**

*Proof.* We first prove the claimed rate of $\varepsilon_{A,k+1} := \frac{Q_A}{\beta_k^{\frac{1}{\nu}}}$, and then that of $\varepsilon_{\varphi,k+1} := \frac{Q_f}{\beta_k}$. From Algorithm 1 step 2 we have that

$$\mathbf{dist}(-\nabla_x \psi_{\beta_k}(x^{k+1}, y^k), \partial g(x^{k+1})) \leq \varepsilon_{k+1}, \quad \forall k \geq 0. \tag{16}$$

By definition of $\psi_\beta$ we therefore have that $\forall k \geq 0$

$$\mathbf{dist}(-\nabla f(x^{k+1}) - J_A^\top(x^{k+1})y^k - \beta_k J_A^\top(x^{k+1})\nabla\phi(A(x^{k+1})), \partial g(x^{k+1})) \leq \varepsilon_{k+1}$$

which yields, after application of the triangle inequality,

$$\mathbf{dist}(-\beta_k J_A^\top(x^{k+1})\nabla\phi(A(x^{k+1})), \partial g(x^{k+1})) \leq \|\nabla f(x^{k+1})\| + \|J_A^\top(x^{k+1})y^k\| + \varepsilon_{k+1}. \tag{17}$$

By (Rockafellar, 1970, Theorem 25.6) it follows that for all $x \in \mathbb{R}^n$,

$$\partial g(x) = \mathrm{cl}(\mathrm{conv}\, S(x)) + N_{\mathrm{dom}\, g}(x),$$

where $S(x)$ is the set of all limits of sequences of the form $\nabla g(x_1), \nabla g(x_2), \ldots$ such that $g$ is differentiable at $x_i$ and $x_i \to x$, and $\mathrm{cl}(\mathrm{conv}\, S(x))$ is the closure of the convex hull of $S(x)$. By Assumption 2 the function $g$ is $G$-Lipschitz continuous on $\mathcal{S}$, and therefore any $v \in \mathrm{cl}(\mathrm{conv}\, S(x))$ satisfies $\|v\| \leq G$.

By again applying the triangle inequality to (17), we obtain

$$\mathbf{dist}(-\beta_k J_A^\top(x^{k+1})\nabla\phi(A(x^{k+1})), N_{\mathrm{dom}\, g}(x^{k+1})) \leq \|\nabla f(x^{k+1})\| + G + \|J_A^\top(x^{k+1})y^k\| + \varepsilon_{k+1}.$$

Since $\nabla\phi(A(x^{k+1})) = \|A(x^{k+1})\|^{\nu-1}A(x^{k+1})$, we can further lower bound the l.h.s. of the previous inequality. We have that $\mathbf{dist}(\alpha x, N_{\mathrm{dom}\, g}(x^{k+1})) = \min_{z \in N_{\mathrm{dom}\, g}(x^{k+1})} \|\alpha x - z\| = \alpha \min_{z \in N_{\mathrm{dom}\, g}(x^{k+1})} \|x - \frac{z}{\alpha}\|$ for $\alpha > 0$ and thus since $N_{\mathrm{dom}\, g}(x^{k+1})$ is a cone, using Assumption 4 we obtain

$$\|\nabla f(x^{k+1})\| + G + \|J_A^\top(x^{k+1})y^k\| + \varepsilon_{k+1} \geq \beta_k\|A(x^{k+1})\|^{\nu-1}\mathbf{dist}(-J_A^\top(x^{k+1})A(x^{k+1}), N_{\mathrm{dom}\, g}(x^{k+1}))$$
$$\geq R\beta_k\|A(x^{k+1})\|^\nu.$$

Therefore, we have the following inequality

$$\|A(x^{k+1})\|^\nu \leq \frac{\|\nabla f(x^{k+1})\| + G + \|J_A^\top(x^{k+1})y^k\| + \varepsilon_{k+1}}{R\beta_k}. \tag{18}$$

By Lemma 1, we have that $\|y^k\| \le y_{\max}$ for all $k \ge 0$. Thus, the constraint violation is upper bounded by

$$\|A(x^{k+1})\| \le \left( \frac{\nabla f_{\max} + G + J_{A\max} y_{\max} + \varepsilon_{k+1}}{R\beta_k} \right)^{1/\nu}, \tag{19}$$

where we have used the Cauchy-Schwarz inequality. The claim regarding $\varepsilon_{A,k+1}$ now follows from the fact that $\varepsilon_{k+1} \le \varepsilon_k$. Now for $\varepsilon_{\varphi,k+1}$, we have by the triangle inequality that

$$\mathbf{dist}(-\nabla_x \psi_{\beta_k}(x^{k+1}, y^{k+1}), \partial g(x^{k+1}))$$
$$\le \mathbf{dist}(-\nabla_x \psi_{\beta_k}(x^{k+1}, y^k), \partial g(x^{k+1})) + \|\nabla_x \psi_{\beta_k}(x^{k+1}, y^{k+1}) - \nabla_x \psi_{\beta_k}(x^{k+1}, y^k)\|$$

The first term of the r.h.s. is upper bounded by $\varepsilon_{k+1}$ due to Algorithm 1 step 2, i.e.,

$$\mathbf{dist}(-\nabla_x \psi_{\beta_k}(x^{k+1}, y^k), \partial g(x^{k+1})) \le \varepsilon_{k+1}$$

and for the second term we have that

$$\|\nabla_x \psi_{\beta_k}(x^{k+1}, y^{k+1}) - \nabla_x \psi_{\beta_k}(x^{k+1}, y^k)\| = \|J_A^\top(x^{k+1})(y^{k+1} - y^k)\| \le J_{A\max}\|y^{k+1} - y^k\|$$
$$= J_{A\max}\sigma_{k+1}\|A(x^{k+1})\|^\nu.$$

Therefore, in combination with (18), we obtain

$$\mathbf{dist}(-\nabla_x \psi_{\beta_k}(x^{k+1}, y^{k+1}), \partial g(x^{k+1})) \le \varepsilon_{k+1} + J_{A\max}\sigma_{k+1} \frac{\nabla f_{\max} + G + J_{A\max} y_{\max} + \varepsilon_{k+1}}{R\beta_k}$$

Thus, since $\sigma_k \le \sigma_1$, $\varepsilon_{k+1} \le \varepsilon_k$ and $\varepsilon_{k+1} = \lambda/\beta_k$, we conclude that $x^{k+1}$ is $(\varepsilon_{\varphi,k+1}, \varepsilon_{A,k+1})$-stationary with multiplier $y^{k+1} + \beta_k \nabla\phi(A(x^{k+1}))$. $\qquad\square$

**Proof of Lemma 3**

*Proof.* Since the gradient of $\psi_\beta$ with respect to its first argument, evaluated at $(x, y)$, is given by

$$\nabla_x \psi_\beta(x, y) = \nabla f(x) + J_A^\top(x)y + \beta J_A^\top(x)\nabla\phi(A(x)), \tag{20}$$

it follows for $x, x' \in \mathcal{S}$ that

$$\|\nabla_x \psi_\beta(x', y) - \nabla_x \psi_\beta(x, y)\| \le \|\nabla f(x') - \nabla f(x)\| + \|J_A(x') - J_A(x)\| \cdot \|y\|$$
$$+ \beta\|J_A^\top(x')\nabla\phi(A(x')) - J_A^\top(x)\nabla\phi(A(x))\|. \tag{21}$$

Using Assumption 1 and Lemma 1, the first two terms are bounded by

$$\|\nabla f(x') - \nabla f(x)\| \le H_f\|x' - x\|^{\nu_f}, \tag{22}$$
$$\|J_A(x') - J_A(x)\| \cdot \|y\| \le H_A\|y\| \cdot \|x' - x\|^{\nu_A}, \tag{23}$$

As for the third term in (21), we have that

$$\beta\|J_A^\top(x')\nabla\phi(A(x')) - J_A^\top(x)\nabla\phi(A(x))\|$$
$$\le \beta\|J_A^\top(x')\nabla\phi(A(x')) - J_A^\top(x)\nabla\phi(A(x'))\| + \beta\|J_A^\top(x)\nabla\phi(A(x')) - J_A^\top(x)\nabla\phi(A(x))\|$$
$$\le \beta\|A(x')\|^\nu \|J_A(x') - J_A(x)\| + \beta\|J_A(x)\|\|\nabla\phi(A(x')) - \nabla\phi(A(x))\|$$
$$\le \beta H_A\|A(x')\|^\nu\|x' - x\|^{\nu_A} + \beta J_{A\max}\|\nabla\phi(A(x')) - \nabla\phi(A(x))\|, \tag{24}$$

where we have used consecutively the triangle inequality, the fact that $\|\nabla\phi(A(x'))\| = \|A(x')\|^\nu$, Assumption 1 and the boundedness of $\|J_A\|$ on $\mathcal{S}$. In light of (Rodomanov & Nesterov, 2020, Theorem 6.3), the function $\phi$ is $(2^{1-\nu}, \nu)$-Hölder smooth. Hence, it follows that

$$\|\nabla\phi(A(x')) - \nabla\phi(A(x))\| \le 2^{1-\nu}\|A(x') - A(x)\|^\nu \le 2^{1-\nu}J_{A\max}^\nu\|x' - x\|^\nu$$

where in the last inequality we also used the Lipschitz continuity of $A$ on $\mathcal{S}$. Putting this back into (24), we obtain

$$\beta\|J_A^\top(x')\nabla\phi(A(x')) - J_A^\top(x)\nabla\phi(A(x))\| \leq \beta H_A A_{\max}^\nu\|x'-x\|^{\nu_A} + 2^{1-\nu}\beta J_{A\max}^{1+\nu}\|x'-x\|^\nu \tag{25}$$

Finally, the claim follows by summing the previous inequalities and using the fact that

$$\|x'-x\|^{\nu_f} = \|x'-x\|^{\nu_f - q}\cdot\|x'-x\|^q \leq D^{\nu_f - q}\|x'-x\|^q \leq \max\left\{1, D^{1-q}\right\}\|x'-x\|^q$$
$$\|x'-x\|^{\nu_A} = \|x'-x\|^{\nu_A - q}\cdot\|x'-x\|^q \leq D^{\nu_A - q}\|x'-x\|^q \leq \max\left\{1, D^{1-q}\right\}\|x'-x\|^q$$
$$\|x'-x\|^\nu = \|x'-x\|^{\nu - q}\cdot\|x'-x\|^q \leq D^{\nu - q}\|x'-x\|^q \leq \max\left\{1, D^{1-q}\right\}\|x'-x\|^q.$$

$\square$

**Proof of Lemma 4**

*Proof.* We first remark that UPFAG monotonically decreases the objective, and hence by Lemma 2 the iterates remain in a compact set. Let $\{\bar{x}_i^{\mathrm{ag}}\}_{i\in\mathbb{N}}$ and $\{x_i^{\mathrm{ag}}\}_{i\in\mathbb{N}}$ be the sequences of iterates generated by the UPFAG method described in (Ghadimi et al., 2019, Algorithm 2) when applied to $\psi_k$. Let $\{\gamma_i\}_{i\in\mathbb{N}}$ be the sequence of stepsizes defined in (Ghadimi et al., 2019, Equation (3.6)). Then in light of (Ghadimi et al., 2019, Equation (3.33)), after discarding some constants, we need $O\left(H_{\beta_k}^{1/q}\left[\frac{L_k(x^k)-L_k^\star}{\varepsilon_{k+1}^2}\right]^{\frac{1+q}{2q}}\right)$ inner iterations to obtain a point $\|\bar{x}_i^{\mathrm{ag}} - x_{i-1}^{\mathrm{ag}}\|/\gamma_i \leq \frac{\varepsilon_{k+1}}{2}$. Now, note that from (Ghadimi et al., 2019, Equation (3.8)) we have that

$$\bar{x}_i^{\mathrm{ag}} = \arg\min_{u\in\mathbb{R}^n}\langle u, \nabla\psi_k(x_{i-1}^{\mathrm{ag}})\rangle + g(u) + \frac{1}{2\gamma_i}\|u - x_{i-1}^{\mathrm{ag}}\|^2$$

and from the optimality conditions for this minimization problem (Rockafellar & Wets, 1998, Theorem 6.12):

$$-\nabla\psi_k(x_{i-1}^{\mathrm{ag}}) - \frac{1}{\gamma_i}(\bar{x}_i^{\mathrm{ag}} - x_{i-1}^{\mathrm{ag}}) \in \partial g(\bar{x}_i^{\mathrm{ag}})$$

By adding and subtracting $\nabla\psi_k(\bar{x}_i^{\mathrm{ag}})$ we further have:

$$-\nabla\psi_k(\bar{x}_i^{\mathrm{ag}}) + \nabla\psi_k(\bar{x}_i^{\mathrm{ag}}) - \nabla\psi_k(x_{i-1}^{\mathrm{ag}}) - \frac{1}{\gamma_i}(\bar{x}_i^{\mathrm{ag}} - x_{i-1}^{\mathrm{ag}}) \in \partial g(\bar{x}_i^{\mathrm{ag}})$$

Therefore, $\mathbf{dist}(-\nabla\psi_k(\bar{x}_i^{\mathrm{ag}}) + \nabla\psi_k(\bar{x}_i^{\mathrm{ag}}) - \nabla\psi_k(x_{i-1}^{\mathrm{ag}}) - \frac{1}{\gamma_i}(\bar{x}_i^{\mathrm{ag}} - x_{i-1}^{\mathrm{ag}}), \partial g(\bar{x}_i^{\mathrm{ag}})) = 0$ and from the triangle inequality

$$\mathbf{dist}(-\nabla\psi_k(\bar{x}_i^{\mathrm{ag}}), \partial g(\bar{x}_i^{\mathrm{ag}})) \leq \|\nabla\psi_k(\bar{x}_i^{\mathrm{ag}}) - \nabla\psi_k(x_{i-1}^{\mathrm{ag}})\| + \frac{1}{\gamma_i}\|\bar{x}_i^{\mathrm{ag}} - x_{i-1}^{\mathrm{ag}}\|$$
$$\leq H_{\beta_k}\|\bar{x}_i^{\mathrm{ag}} - x_{i-1}^{\mathrm{ag}}\|^q + \frac{\varepsilon_{k+1}}{2} \leq H_{\beta_k}\frac{\varepsilon_{k+1}^q\gamma_i^q}{2^q} + \frac{\varepsilon_{k+1}}{2},$$

where in the third inequality we used the bound $\|\bar{x}_i^{\mathrm{ag}} - x_{i-1}^{\mathrm{ag}}\|/\gamma_i \leq \frac{\varepsilon_{k+1}}{2}$ and the Hölder continuity of $\nabla\psi_k$, Lemma 3. Therefore, by choosing $\gamma_i \leq \frac{\varepsilon_{k+1}^{(1-q)/q}}{(2^{1-q}H_{\beta_k})^{1/q}}$ we obtain the claimed result. $\square$

**Proof of Theorem 2**

*Proof.* Remark that by Lemma 2(ii) the iterates remain in a compact set. Let us start by defining the *first* power ALM (outer) iteration $K_A$ for which

$$\varepsilon_A \geq \varepsilon_{A, K_A + 1} = \left(\frac{\nabla f_{\max} + G + J_{A\max}y_{\max} + \varepsilon_1}{R\beta_{K_A}}\right)^{\frac{1}{\nu}} := \frac{Q_A}{\beta_{K_A}^{1/\nu}}. \tag{26}$$

or, equivalently, for which $\beta_{K_A} \geq \frac{Q_A}{\epsilon_A^\nu}$. Since $K_A$ is the smallest iteration index for which this holds and $\beta_k$ is increasing, it follows that $\beta_{K_A - 1} < \frac{Q_A}{\varepsilon_A^\nu}$. It follows from

$$\|A(x^{K_A+1})\| \leq \left(\frac{\nabla f_{\max} + G + J_{A\max}y_{\max} + \varepsilon_1}{R\beta_{K_A}}\right)^{\frac{1}{\nu}} \tag{27}$$

that also $\varepsilon_A \geq \|A(x^{K_A+1})\|$ holds. From the update rule for $\beta_k$ we have that

$$\omega^{K_A-2} < \frac{1}{\varepsilon_A^\nu}\left(\frac{Q_A}{\beta_1}\right) := \frac{Q_A'}{\varepsilon_A^\nu}. \tag{28}$$

After taking the logarithm of both sides, we obtain that

$$K_A = \left\lceil \log_\omega\left(\frac{Q_A'}{\varepsilon_A^\nu}\right)\right\rceil + 2. \tag{29}$$

Now, the number of total UPFAG iterations needed to obtain a point $x^{K_A+1}$ satisfying (27) is upper bounded by the sum of the calls to UPFAG for all outer (power ALM) iterations. Note that since $\beta_k$ is increasing geometrically, the Hölder smoothness modulus defined in Lemma 3 is determined by it for large enough $k$. Therefore, in light of Lemma 4 we require at most $T_A$ UPFAG iterations, where

$$
\begin{aligned}
T_A &= \sum_{k=1}^{K_A} O\left(H_{\beta_k}^{1/q}\left[\frac{L_k(x^k) - L_k^\star}{\varepsilon_{k+1}^2}\right]^{\frac{1+q}{2q}}\right) = \sum_{k=1}^{K_A} O\left(\beta_k^{1/q}\left[D^{q+1}\beta_k^3\right]^{\frac{1+q}{2q}}\right) \\
&= O\left(K_A\beta_{K_A}^{1/q}\left[D^{q+1}\beta_{K_A}^3\right]^{\frac{1+q}{2q}}\right) \leq O\left(K_A\frac{(\omega Q_A)^{1/q}}{\varepsilon_A^{\nu/q}}\left[D^{q+1}\left(\omega\frac{Q_A}{\varepsilon_A^\nu}\right)^3\right]^{\frac{1+q}{2q}}\right) \\
&= O\left(K_A\frac{Q_A^{1/q}}{\varepsilon_A^{\nu/q}}D^{\frac{(q+1)^2}{2q}}\frac{Q_A^{3\frac{1+q}{2q}}}{\varepsilon_A^{3\nu\frac{1+q}{2q}}}\right) = O\left(\frac{K_A Q_A^{\frac{5+3q}{2q}}D^{\frac{(1+q)^2}{2q}}}{\varepsilon_A^{\frac{\nu}{q}+\frac{3\nu(1+q)}{2}}}\right),
\end{aligned}
$$

where the first equality follows by Lemma 3 and by the fact that $\varepsilon_{k+1} = \lambda/\beta_k$. The second equality follows by unrolling the sum and the first inequality by the fact that $\beta_k = \omega\beta_{k-1} \leq \omega\frac{Q_A}{\varepsilon_A^\nu}$. Thus, we obtain

$$T_A = O\left(\frac{\log_\omega\left(\frac{Q_A'}{\varepsilon_A^\nu}\right)Q_A^{\frac{5+3q}{2q}}D^{\frac{(1+q)^2}{2q}}}{\varepsilon_A^{\frac{\nu}{q}+\frac{3\nu(1+q)}{2}}}\right) = \tilde{O}\left(\frac{Q_A^{\frac{5+3q}{2q}}D^{\frac{(1+q)^2}{2q}}}{\varepsilon_A^{\frac{\nu}{q}+\frac{3\nu(1+q)}{2}}}\right).$$

Regarding the suboptimality tolerance, we can in a similar way define the *first* power ALM (outer) iteration $K_\varphi$ for which

$$\varepsilon_\varphi \geq \varepsilon_{\varphi,K_\varphi+1} = \frac{1}{\beta_{K_\varphi}}\left(\lambda + J_{A\max}\sigma_1\frac{\nabla f_{\max} + G + J_{A\max}y_{\max} + \varepsilon_1}{R}\right) := \frac{Q_f}{\beta_{K_\varphi}} \tag{30}$$

or, equivalently, $\beta_{K_\varphi} \geq \frac{Q_f}{\varepsilon_\varphi}$. Since $K_\varphi$ is the first iteration for which this holds, it follows that $\beta_{K_\varphi-1} < \frac{Q_f}{\varepsilon_\varphi}$. From the update rule for $\beta_k$ we have that

$$\omega^{K_\varphi-2} < \frac{1}{\varepsilon_\varphi}\left(\frac{Q_f}{\beta_1}\right) := \frac{Q_f'}{\varepsilon_\varphi} \tag{31}$$

After taking the logarithm on both sides, we obtain

$$K_\varphi = \left\lceil \log_\omega\left(\frac{Q_f'}{\varepsilon_\varphi}\right)\right\rceil + 2. \tag{32}$$

Hence, we require at most $T_\varphi$ UPFAG iterations to obtain a point $x^{K_\varphi+1}$ satisfying (30), where

$$
\begin{aligned}
T_\varphi &= \sum_{k=1}^{K_\varphi} O\left( H_{\beta_k}^{1/q} \left[ \frac{L_k(x^k) - L_k^\star}{\varepsilon_{k+1}^2} \right]^{\frac{1+q}{2q}} \right) = \sum_{k=1}^{K_\varphi} O\left( \beta_k^{1/q} \left[ D^{q+1} \beta_k^3 \right]^{\frac{1+q}{2q}} \right) \\
&= O\left( K_\varphi \beta_{K_\varphi}^{1/q} \left[ D^{q+1} \beta_{K_\varphi}^3 \right]^{\frac{1+q}{2q}} \right) \leq O\left( K_\varphi \frac{Q_f^{1/q}}{\varepsilon_\varphi^{1/q}} \left[ D^{q+1} \left( \omega \frac{Q_f}{\varepsilon_\varphi} \right)^3 \right]^{\frac{1+q}{2q}} \right) \\
&= O\left( K_\varphi \frac{Q_f^{1/q}}{\varepsilon_\varphi^{1/q}} D^{\frac{(q+1)^2}{2q}} \frac{Q_f^{\frac{3(q+1)}{2q}}}{\varepsilon_\varphi^{\frac{3(q+1)}{2q}}} \right) = O\left( \frac{K_\varphi Q_f^{\frac{5+3q}{2q}} D^{\frac{(1+q)^2}{2q}}}{\varepsilon_\varphi^{\frac{5+3q}{2q}}} \right),
\end{aligned}
$$

and the sequence of bounds follows from similar arguments as for $T_A$. Thus, we obtain

$$
T_\varphi = O\left( \frac{\log_\omega \left( \frac{Q_f'}{\varepsilon_\varphi} \right) Q_f^{\frac{5+3q}{2q}} D^{\frac{(1+q)^2}{2q}}}{\varepsilon_\varphi^{\frac{5+3q}{2q}}} \right) = \tilde{O}\left( \frac{Q_f^{\frac{5+3q}{2q}} D^{\frac{(1+q)^2}{2q}}}{\varepsilon_\varphi^{\frac{5+3q}{2q}}} \right).
$$

$\square$

## A.2 Proofs of section 3

**Proof of Lemma 5**

*Proof.* The function $f$ is $L_f$-weakly convex, since it is $L_f$-Lipschitz smooth, i.e. for any $x, x' \in \mathcal{X}$ the following inequality holds:

$$
f(x) \geq f(x') + \langle \nabla f(x'), x - x' \rangle - \frac{L_f}{2} \|x - x'\|^2 \tag{33}
$$

Moreover, following the proof of (Drusvyatskiy & Paquette, 2019, Lemma 4.2), we have for any $x, x' \in \mathcal{X}$, and $y \in \mathbb{R}^m$ that

$$
\begin{aligned}
\langle y, A(x') \rangle &= \langle y, A(x) \rangle + \langle y, A(x') - A(x) \rangle \\
&\geq \langle y, A(x) \rangle + \langle y, J_A(x)(x' - x) \rangle - \frac{L_A \|y\|}{2} \|x' - x\|^2 \\
&= \langle y, A(x) \rangle + \langle J_A^\top(x)y, (x' - x) \rangle - \frac{L_A \|y\|}{2} \|x' - x\|^2, \tag{34}
\end{aligned}
$$

where the inequality follows from the Lipschitz-continuity of $J_A$, i.e., from the fact that $\|A(x') - A(x) - J_A(x)(x' - x)\| \leq \frac{L_A}{2}\|x' - x\|^2$. Likewise, for any $x, x' \in \mathcal{X}$ we have that

$$
\begin{aligned}
\beta\phi(A(x)) &\geq \beta\phi(A(x')) + \langle \nabla\phi(A(x')), A(x) - A(x') \rangle \\
&\geq \beta\phi(A(x')) + \langle \nabla\phi(A(x')), J_A(x')(x - x') \rangle - \frac{L_A \|\nabla\phi(A(x'))\|}{2} \|x - x'\|^2 \\
&\geq \beta\phi(A(x')) + \beta\langle J_A^\top(x')\nabla\phi(A(x')), x - x' \rangle - \frac{\beta L_A A_{\max}^\nu}{2} \|x - x'\|^2 \tag{35}
\end{aligned}
$$

by consecutively exploiting convexity of $\phi$ between points $A(x)$ and $A(x')$, Lipschitz-smoothness of $A$ and the fact that $\|\nabla\phi(A(x))\| = \|A(x)\|^\nu \leq A_{\max}^\nu$. Summing up (33), (34) and (35) we obtain:

$$
L_\beta(x, y) \geq L_\beta(x', y) + \langle \nabla_x L_\beta(x', y), x - x' \rangle - \frac{L_f + L_A \|y\| + \beta L_A A_{\max}^\nu}{2} \|x - x'\|^2
$$

These observations directly imply that for any $y \in \mathbb{R}^m$ the augmented Lagrangian is $\rho$-weakly convex on $\mathcal{X}$ in its first argument, with

$$
\rho := L_f + L_A(\|y\| + \beta A_{\max}^\nu).
$$

$\square$

**Proof of Theorem 3**

The proof requires the following series of lemmata. Note that $F$ is defined in Algorithm 2 step 2 and is a $(H_F, \nu)$-Hölder smooth and $\rho$-strongly convex function.

**Lemma 7.** *Let $x \in \mathcal{X}$ and define the point $x^+ := \arg\min_{u \in \mathcal{X}} \langle \nabla F(x), u - x \rangle + \frac{1}{2\gamma} \|u - x\|^2$ with $\gamma = \frac{1}{H_F} \varepsilon^{\frac{1-\nu}{1+\nu}}$, $\varepsilon > 0$. Then,*

$$\frac{1}{2\gamma} \|x^+ - x\|^2 \leq F(x) - F(x^+) + \frac{H_F}{2} \varepsilon. \tag{36}$$

*Proof.* The inner problem in the definition of $x^+$ is $\frac{1}{\gamma}$-strongly convex and as such for any $u \in \mathcal{X}$ we obtain

$$\langle \nabla F(x), x^+ - x \rangle + \frac{1}{2\gamma} \|x^+ - x\|^2 \leq \langle \nabla F(x), x - u \rangle + \frac{1}{2\gamma} \|x - u\|^2 - \frac{1}{2\gamma} \|x^+ - u\|^2.$$

By choosing $u = x \in \mathcal{X}$ we get

$$0 \geq \langle \nabla F(x), x^+ - x \rangle + \frac{1}{\gamma} \|x^+ - x\|^2 \tag{37}$$

In light of Young's inequality we have for any $a, b \geq 0$ that $ab \leq \frac{a^{p_1}}{p_1} + \frac{b^{p_2}}{p_2}$ with $\frac{1}{p_1} + \frac{1}{p_2} = 1$. Choosing $p_1 = \frac{2}{1+\nu}, p_2 = \frac{2}{1-\nu}$, it follows for $a = \varepsilon^{-\frac{1-\nu}{2}}$, $b = \varepsilon^{\frac{1-\nu}{2}}$ that

$$\|x^+ - x\|^{\nu+1} \leq \frac{1+\nu}{2} \varepsilon^{-\frac{1-\nu}{1+\nu}} \|x^+ - x\|^2 + \frac{1-\nu}{2} \varepsilon. \tag{38}$$

Note that for $\nu = 1$ the inequality becomes an equality and thus the case $\nu = 1$ is also covered. Therefore, from the Hölder smoothness inequality for $F$ between points $x, x^+ \in \mathcal{X}$ we get:

$$\begin{aligned} F(x^+) &\leq F(x) + \langle \nabla F(x), x^+ - x \rangle + \frac{H_F}{\nu+1} \|x^+ - x\|^{\nu+1} \\ &\leq F(x) + \langle \nabla F(x), x^+ - x \rangle + \frac{H_F}{2} \varepsilon^{-\frac{1-\nu}{1+\nu}} \|x^+ - x\|^2 + \frac{1-\nu}{1+\nu} \frac{H_F}{2} \varepsilon \end{aligned} \tag{39}$$

Substituting (39) in (37) we obtain:

$$0 \geq F(x^+) - F(x) + \frac{1}{2\gamma} \|x^+ - x\|^2 - \frac{1-\nu}{1+\nu} \frac{H_F}{2} \varepsilon$$

which proves the claim. $\qquad \square$

**Lemma 8.** *Suppose that a point $x \in \mathcal{X}$ satisfies $F(x) - F(x^\star) \leq \varepsilon$ for some $\varepsilon > 0$. Let $\gamma = \frac{1}{H_F} \varepsilon^{\frac{1-\nu}{1+\nu}}$ and $x^+ = \arg\min_{u \in \mathcal{X}} \langle \nabla F(x), u - x \rangle + \frac{1}{2\gamma} \|u - x\|^2$. Then, $\frac{\|x^+ - x\|^2}{\gamma^2} \leq 2H_F(1 + H_F) \varepsilon^{\frac{2\nu}{1+\nu}}$.*

*Proof.* From Lemma 7 and $F(x) - F(x^+) \leq F(x) - F(x^\star) \leq \varepsilon$ it follows that

$$\frac{\|x^+ - x\|^2}{\gamma^2} \leq \frac{2}{\gamma} \left( \varepsilon + \frac{H_F}{2} \varepsilon \right) \tag{40}$$

$$= 2H_F \varepsilon^{-\frac{1-\nu}{1+\nu}} \left( \varepsilon + \frac{H_F}{2} \varepsilon \right). \tag{41}$$

Therefore, we obtain

$$\frac{\|x^+ - x\|^2}{\gamma^2} \leq 2H_F(1 + H_F) \varepsilon^{\frac{2\nu}{1+\nu}}.$$

$\qquad \square$

**Lemma 9.** *Suppose that a point $x \in \mathcal{X}$ satisfies $F(x) - F(x^\star) \leq \varepsilon$ for some $\varepsilon > 0$. Let $\gamma = \frac{1}{H_F} \varepsilon^{\frac{1-\nu}{1+\nu}}$ and $x^+ = \arg\min_{u \in \mathcal{X}} \langle \nabla F(x), u - x \rangle + \frac{1}{2\gamma} \|u - x\|^2$. Then,*

$$\mathbf{dist}(-\nabla F(x^+), N_{\mathcal{X}}(x^+)) \leq \left( H_F^{1-\nu} (2H_F(1 + H_F))^{\frac{\nu}{2}} + (2H_F(1 + H_F))^{1/2} \right) \varepsilon^{\frac{\nu}{1+\nu}}.$$

*Proof.* By the optimality conditions of the inner minimization in $x^+ = \arg\min_{u \in \mathcal{X}} \langle \nabla F(x), u - x \rangle + \frac{1}{2\gamma} \|u - x\|^2$, we have that

$$-\nabla F(x) - \tfrac{1}{\gamma}(x^+ - x) \in N_{\mathcal{X}}(x^+),$$

which implies that

$$-\nabla F(x^+) + \left(\nabla F(x^+) - \nabla F(x) - \tfrac{1}{\gamma}(x^+ - x)\right) \in N_{\mathcal{X}}(x^+).$$

Observe that from the triangle inequality and the Hölder smoothness of $F$, we have

$$\|\nabla F(x^+) - \nabla F(x) - \tfrac{1}{\gamma}(x^+ - x)\| \leq H_F \|x^+ - x\|^\nu + \tfrac{1}{\gamma}\|x^+ - x\| \tag{42}$$

and by Lemma 8 we can further bound

$$
\begin{aligned}
H_F \|x^+ - x\|^\nu + \tfrac{1}{\gamma}\|x^+ - x\| &\leq \gamma^\nu H_F \frac{\|x^+ - x\|^\nu}{\gamma^\nu} + \tfrac{1}{\gamma}\|x^+ - x\| \\
&= \gamma^\nu H_F \left[(2H_F(1+H_F))^{\frac{\nu}{2}} \varepsilon^{\frac{\nu^2}{1+\nu}}\right] + (2H_F(1+H_F))^{1/2}\varepsilon^{\frac{\nu}{1+\nu}} \\
&= \left[(\tfrac{1}{H_F})^\nu \varepsilon^{\frac{\nu(1-\nu)}{1+\nu}}\right] H_F \left[(2H_F(1+H_F))^{\frac{\nu}{2}} \varepsilon^{\frac{\nu^2}{1+\nu}}\right] + (2H_F(1+H_F))^{1/2}\varepsilon^{\frac{\nu}{1+\nu}} \\
&= \varepsilon^{\frac{\nu}{\nu+1}}\left(H_F^{1-\nu}(2H_F(1+H_F))^{\frac{\nu}{2}} + (2H_F(1+H_F))^{1/2}\right)
\end{aligned}
$$

The result then follows by the fact that $\mathbf{dist}(-\nabla F(x^+), N_{\mathcal{X}}(x^+)) \leq \mathbf{dist}(\nabla F(x^+) - \nabla F(x) - \tfrac{1}{\gamma}(x^+ - x), N_{\mathcal{X}}(x^+)) \leq \|\nabla F(x^+) - \nabla F(x) - \tfrac{1}{\gamma}(x^+ - x)\|$. $\square$

*Proof.* Consider a point $x \in \mathcal{X}$ such that

$$F(x) - F(x^\star) \leq \bar{\varepsilon} := \left[\overline{H}^{-1}\varepsilon\right]^{\frac{1+\nu}{\nu}}.$$

Then, it follows from Lemma 9 that for $\gamma = \frac{1}{H_F}\bar{\varepsilon}^{\frac{1-\nu}{1+\nu}} = \frac{1}{H_F}\left[\overline{H}^{-1}\right]^{\frac{1-\nu}{\nu}}\varepsilon^{\frac{1-\nu}{\nu}}$ the point $x^+$ satisfies

$$\mathbf{dist}(-\nabla F(x^+), N_{\mathcal{X}}(x^+)) \leq \bar{H}\bar{\varepsilon}^{\frac{\nu}{1+\nu}} = \varepsilon.$$

Thus, in light of (Devolder et al., 2014, §6.2 ), it takes at most $T$ iterations of the FGM, and a single proximal-gradient step with stepsize $\gamma$ to find such a point $x^+$, where

$$T = \widetilde{O}\left(\frac{H_F^{\frac{2}{1+3\nu}}}{\rho^{\frac{\nu+1}{3\nu+1}}} \bar{\varepsilon}^{-\frac{1-\nu}{1+3\nu}}\right) \tag{43}$$

$$= \widetilde{O}\left(\frac{H_F^{\frac{2}{1+3\nu}}}{\rho^{\frac{\nu+1}{3\nu+1}}}\left(\overline{H}^{-\frac{1+\nu}{\nu}}\varepsilon^{\frac{1+\nu}{\nu}}\right)^{-\frac{1-\nu}{1+3\nu}}\right) \tag{44}$$

$$= \widetilde{O}\left(\frac{H_F^{\frac{2}{1+3\nu}}}{\rho^{\frac{\nu+1}{3\nu+1}}}\overline{H}^{\frac{1+\nu}{\nu}\frac{1-\nu}{1+3\nu}}\varepsilon^{-\frac{1+\nu}{\nu}\frac{1-\nu}{1+3\nu}}\right). \tag{45}$$

$\square$

## Proof of Theorem 4

*Proof.* Let $F_k(x) := \psi(x) + \rho\|x - x^k\|^2$. Since $\psi$ is $\rho$-weakly convex and has $(H, \nu)$-Hölder continuous gradients, it follows that $F_k$ is $\rho$-strongly convex and has $(H_F, \nu)$-Hölder continuous gradients on $\mathcal{X}$, with $H_F = H + \rho \max\left\{1, D^{1-\nu}\right\}$.

We can now trace the steps of (Li et al., 2021, Theorem 1) to conclude that Algorithm 2 terminates after at most

$$T = \lceil \frac{32\rho}{\varepsilon^2}(\psi(x^1) - \psi^\star) + 1\rceil$$

iterations and that the output $x$ satisfies $\mathbf{dist}(-\nabla\psi(x), N_{\mathcal{X}}(x)) \leq \varepsilon$.

Denote henceforth $F_k^\star = \min_{x \in \mathcal{X}} F_k(x)$. By Algorithm 2 step 3 and (Rockafellar & Wets, 1998, Theorem 10.1) we have that $\mathbf{dist}(0, \partial(F_k+\delta_{\mathcal{X}})(x^{k+1})) = \mathbf{dist}(-\nabla F_k(x^{k+1}), N_{\mathcal{X}}(x^{k+1})) \leq \frac{\varepsilon}{4}$, and by $\rho$-strong convexity of $F_k$ and convexity of $\delta_{\mathcal{X}}$ we have that

$$F_k^\star \geq F_k(x^{k+1}) + \langle v^{k+1}, x^\star - x^{k+1}\rangle + \frac{\rho}{2}\|x^{k+1} - x^\star\|^2, \qquad v^{k+1} \in \partial(F_k + \delta_{\mathcal{X}})(x^{k+1}).$$

Combining these two inequalities yields $F_k(x^{k+1}) - F_k^\star \leq \frac{\varepsilon^2}{32\rho}$ and hence we obtain $\psi(x^{k+1}) + \rho\|x^{k+1} - x^k\|^2 - \psi(x^k) \leq \frac{\varepsilon^2}{32\rho}$. Thus,

$$\psi(x^T) - \psi(x^1) + \rho\sum_{k=1}^{T-1}\|x^{k+1} - x^k\|^2 \leq (T-1)\frac{\varepsilon^2}{32\rho}$$

$$(T-1)\min_{1\leq k\leq T-1}\|x^{k+1} - x^k\|^2 \leq \frac{1}{\rho}\left((T-1)\frac{\varepsilon^2}{32\rho} + \left[\psi(x^1) - \psi(x^T)\right]\right)$$

$$2\rho\min_{1\leq k\leq T-1}\|x^{k+1} - x^k\| \leq 2\sqrt{\frac{\varepsilon^2}{32} + \frac{\rho\left[\psi(x^1) - \psi^\star\right]}{T-1}}$$

For $T - 1 \geq \frac{32\rho}{\varepsilon^2}(\psi(x^1) - \psi^\star)$ this yields

$$2\rho\min_{1\leq k\leq T}\|x^{k+1} - x^k\| \leq \frac{\varepsilon}{2},$$

meaning that Algorithm 2 must have terminated. Denote by $x^T$ the iterate returned by Algorithm 2 upon termination. Since $2\rho\|x^T - x^{T-1}\| \leq \frac{\varepsilon}{2}$ and $\mathbf{dist}(0, \partial(F_{T-1} + \delta_{\mathcal{X}})(x^T)) \leq \frac{\varepsilon}{4}$, we conclude that

$$\mathbf{dist}(0, \partial(\psi + \delta_{\mathcal{X}}(x^T))) \leq \mathbf{dist}(0, \partial(F_{T-1} + \delta_{\mathcal{X}}(x^T))) + 2\rho\|x^T - x^{T-1}\| \leq \varepsilon.$$

This concludes the proof, since $\partial(\psi + \delta_{\mathcal{X}})(\cdot) = \nabla\psi(\cdot) + N_{\mathcal{X}}(\cdot)$. $\qquad\square$

**Proof of Theorem 5**

*Proof.* We denote the $t$-th iterate of Algorithm 2 (iPPM) within the $k$-th power ALM outer iteration by $x_k^t$. At $x_k^t$, Nesterov's FGM is used to minimize $F_k^t := L_{\beta_k}(\cdot, y^k) + \rho_k\|\cdot - x_k^t\|^2$, which is $\rho_k$-strongly convex and $(H_{F_k}, \nu)$-Hölder smooth with $H_{F_k} = H_k + \rho_k\max\{1, D^{1-\nu}\}$. In light of Theorem 3, we require at most $T_k^{FGM}$ iterations to find an $\frac{\varepsilon_{k+1}}{4}$ stationary point of $F_k^t + \delta_{\mathcal{X}}$, with

$$T_k^{FGM} = \widetilde{O}\left(\frac{H_{F_k}^{\frac{2}{1+3\nu}}}{\rho_k^{\frac{1+\nu}{1+3\nu}}}\overline{H}_k^{\frac{1+\nu}{\nu}\frac{1-\nu}{1+3\nu}}\varepsilon_{k+1}^{-\frac{1+\nu}{\nu}\frac{1-\nu}{1+3\nu}}\right), \tag{46}$$

where $\overline{H}_k = H_{F_k}^{1-\nu}(2H_{F_k}(1 + H_{F_k}))^{\frac{\nu}{2}} + (2H_{F_k}(1 + H_{F_k}))^{1/2}$. Remark that $T_k^{FGM}$ is independent of $t$. Since $\rho_k = O(\beta_k), H_{F_k} = O(\beta_k), \bar{H}_k = O(\beta_k)$, this expression simplifies to

$$T_k^{FGM} = \widetilde{O}\left(\beta_k^{\frac{1-\nu}{1+3\nu}\left(1 + \frac{1+\nu}{\nu}\right)}\varepsilon_{k+1}^{-\frac{1+\nu}{\nu}\frac{1-\nu}{1+3\nu}}\right).$$

Likewise, at $x^k$, the iPPM is used to minimize $L_{\beta_k}(\cdot, y^k)$, which is $\rho_k$-weakly convex. By Theorem 4, we require at most $T_k^{PPM}$ iterations of Algorithm 2 to find a point satisfying the update rule of step 2 in Algorithm 1, where

$$T_k^{PPM} = \lceil\frac{32\rho_k}{\varepsilon_{k+1}^2}(\psi_k(x^k) - \psi_k^\star) + 1\rceil \tag{47}$$

From (18) we have that $\|A(x^k)\|^q \leq \left(\frac{\nabla f_{\max} + G + J_{A\max}y_{\max} + \varepsilon_k}{R\beta_{k-1}}\right)$ for all $k \geq 1$ and using $\beta_k = \omega\beta_{k-1}$ we obtain

$$\beta_k \|A(x^k)\|^\nu \leq \omega \frac{\nabla f_{\max} + G + J_{A\max}y_{\max} + \varepsilon_k}{R} \tag{48}$$

and hence we obtain the following bound for $k \geq 2$:

$$
\begin{aligned}
\psi_k(x^k) &= f(x^k) + \langle y^k, A(x^k)\rangle + \frac{\beta_k}{1+\nu}\|A(x^k)\|^{\nu+1} \\
&\leq f_{\max} + y_{\max}\|A(x^k)\| + \frac{\omega}{1+\nu}\left(\frac{\nabla f_{\max} + G + J_{A\max}y_{\max} + \varepsilon_k}{R}\right)\|A(x^k)\| \\
&\leq f_{\max} + y_{\max}\left(\frac{\nabla f_{\max} + G + J_{A\max}y_{\max} + \varepsilon_k}{R\beta_{k-1}}\right)^{\frac{1}{\nu}} \\
&\quad + \frac{\omega}{1+\nu}\left(\frac{\nabla f_{\max} + G + J_{A\max}y_{\max} + \varepsilon_k}{R}\right)\left(\frac{\nabla f_{\max} + G + J_{A\max}y_{\max} + \varepsilon_k}{R\beta_{k-1}}\right)^{\frac{1}{\nu}} \\
&\leq f_{\max} + y_{\max} + \left(\frac{\nabla f_{\max} + G + J_{A\max}y_{\max} + \varepsilon_1}{R\beta_1}\right)^{\frac{1}{\nu}} \\
&\quad + \frac{\omega}{1+\nu}\left(\frac{\nabla f_{\max} + G + J_{A\max}y_{\max} + \varepsilon_1}{R}\right)\left(\frac{\nabla f_{\max} + G + J_{A\max}y_{\max} + \varepsilon_1}{R\beta_1}\right)^{\frac{1}{\nu}} =: C.
\end{aligned}
$$

The first inequality follows by $f$ being continuous on the compact set $\mathcal{X}$, $f(x^k) \leq \max_{x\in\mathcal{X}} f(x) := f_{\max}$, and (48) and the second one by (19). The last inequality follows by $\varepsilon_k < \varepsilon_1$ and $\beta_k > \beta_1$. Moreover, we have for all $x \in \mathcal{X}$

$$\psi_k(x) \geq f(x) + \langle y^k, A(x)\rangle \geq -f_{\max} - y_{\max}A_{\max}$$

and as such we can bound

$$\psi_k(x^k) - \psi^\star \leq C + f_{\max} + y_{\max}A_{\max}.$$

Using this expression and Lemma 5, we can rewrite the number of iPPM iterations as

$$T_k^{PPM} = \widetilde{O}\left(\frac{\beta_k}{\varepsilon_{k+1}^2}\right)$$

We now define the *first* power ALM (outer) iteration $K_A$ for which

$$\varepsilon_A \geq \varepsilon_{A,K_A+1} = \left(\frac{\nabla f_{\max} + G + J_{A\max}y_{\max} + \varepsilon_1}{R\beta_{K_A}}\right)^{\frac{1}{\nu}} := \frac{Q_A}{\beta_{K_A}^{1/\nu}} \tag{49}$$

and from (29) we have that

$$K_A = \left\lceil \log_\omega\left(\frac{Q_A'}{\varepsilon_A^\nu}\right)\right\rceil + 2.$$

Hence, we require at most $T_A$ total FGM iterations to obtain a point $x^{K_A+1}$, where

$$
\begin{aligned}
T_A &= \sum_{k=1}^{K_A} T_k^{PPM} T_k^{FGM} \\
&= \sum_{k=1}^{K_A} \widetilde{O}\left(\frac{\beta_k}{\varepsilon_{k+1}^2}\beta_k^{\frac{1-\nu}{1+3\nu}\left(1+\frac{1+\nu}{\nu}\right)}\varepsilon_{k+1}^{-\frac{1+\nu}{\nu}\frac{1-\nu}{1+3\nu}}\right) = \sum_{k=1}^{K_A} \widetilde{O}\left(\beta_k^{1+\frac{1-\nu}{1+3\nu}\left(1+\frac{1+\nu}{\nu}\right)}\beta_k^{2+\frac{1+\nu}{\nu}\frac{1-\nu}{1+3\nu}}\right) \\
&= \sum_{k=1}^{K_A} \widetilde{O}\left(\beta_k^{3+\frac{1-\nu}{1+3\nu}\left(1+2\frac{1+\nu}{\nu}\right)}\right) \\
&\leq \widetilde{O}\left(K_A\beta_{K_A}^{3+\frac{1-\nu}{1+3\nu}\left(1+2\frac{1+\nu}{\nu}\right)}\right) = \widetilde{O}\left(\beta_{K_A}^{3+\frac{1-\nu}{1+3\nu}\left(1+2\frac{1+\nu}{\nu}\right)}\right)
\end{aligned}
$$

By substitution of $\beta_{K_A} = \omega \beta_{K_A - 1} < \frac{Q_A}{\varepsilon_A^\nu}$ this yields

$$T_A = \widetilde{O}\left(\varepsilon_A^{-3\nu - \frac{1-\nu}{1+3\nu}(3\nu+2)}\right).$$

In a similar way to the proof of Theorem 2 we can define the *first* power ALM (outer) iteration $K_\varphi$ for which

$$\varepsilon_\varphi \geq \varepsilon_{\varphi, K_\varphi + 1} = \frac{1}{\beta_{K_\varphi}}\left(\lambda + J_{A\max}\sigma_1 \frac{\nabla f_{\max} + G + J_{A\max}y_{\max} + \varepsilon_1}{R}\right) := \frac{Q_f}{\beta_{K_\varphi}}$$

and obtain

$$K_\varphi = \left\lceil \log_\omega\left(\frac{Q_f'}{\varepsilon_\varphi}\right)\right\rceil + 2.$$

$$T_\varphi = \widetilde{O}\left(\beta_{K_\varphi}^{3 + \frac{1-\nu}{1+3\nu}\left(1 + 2\frac{1+\nu}{\nu}\right)}\right)$$

By substitution of $\beta_{K_\varphi} = \omega \beta_{K_\varphi - 1} < \frac{Q_f}{\varepsilon_\varphi}$ this yields

$$T_\varphi = \widetilde{O}\left(\varepsilon_\varphi^{-3 - \frac{1-\nu}{1+3\nu}\left(1 + \frac{2(1+\nu)}{\nu}\right)}\right).$$

$\square$

**Proof of Theorem 6**

*Proof.* We denote the $t$-th iterate of Algorithm 2 (iPPM) within the $k$-th power ALM outer iteration by $x_k^t$. At $x_k^t$, Nesterov's FGM is used to minimize $F_k^t := L_{\beta_k}(\cdot, y^k) + \rho_k \|\cdot - x_k^t\|^2$, which is $\rho_k$-strongly convex and $(H_{F_k}, \nu)$-Hölder smooth with $H_{F_k} = H_k + \rho_k \max\{1, D^{1-\nu}\}$. In light of Theorem 3, we require at most $T_k^{FGM}$ iterations to find an $\frac{\varepsilon_{k+1}}{4}$ stationary point of $F_k^t + \delta_\mathcal{X}$, with

$$T_k^{FGM} = \widetilde{O}\left(\frac{H_{F_k}^{\frac{2}{1+3\nu}}}{\rho_k^{\frac{1+\nu}{1+3\nu}}} \overline{H}_k^{\frac{1+\nu}{\nu}\frac{1-\nu}{1+3\nu}} \varepsilon_{k+1}^{-\frac{1+\nu}{\nu}\frac{1-\nu}{1+3\nu}}\right), \tag{50}$$

where $\overline{H}_k = H_{F_k}^{1-\nu}(2H_{F_k}(1 + H_{F_k}))^{\frac{\nu}{2}} + (2H_{F_k}(1 + H_{F_k}))^{1/2}$. Remark that $T_k^{FGM}$ is independent of $t$. Since $\rho_k = O(1), H_{F_k} = O(\beta_k), \overline{H}_k = O(\beta_k)$, this expression simplifies to

$$T_k^{FGM} = \widetilde{O}\left(\beta_k^{\frac{1-\nu}{1+3\nu}\frac{1+\nu}{\nu} + \frac{2}{1+3\nu}} \varepsilon_{k+1}^{-\frac{1+\nu}{\nu}\frac{1-\nu}{1+3\nu}}\right).$$

Likewise, at $x^k$, the iPPM is used to minimize $L_{\beta_k}(\cdot, y^k)$, which is $\rho_k$-weakly convex. By Theorem 4, we require at most $T_k^{PPM}$ iterations of Algorithm 2 to find a point satisfying the update rule of step 2 in Algorithm 1, where

$$T_k^{PPM} = \lceil \frac{32\rho_k}{\varepsilon_{k+1}^2}(\psi_k(x^k) - \psi_k^\star) + 1\rceil \tag{51}$$

Following the same steps as in the proof of Theorem 5, we can bound for some $C > 0$,

$$\psi_k(x^k) - \psi^\star \leq C + f_{\max} + y_{\max}A_{\max}.$$

Using this expression and $\rho_k = O(1)$ (cf. Lemma 5), we can rewrite the number of iPPM iterations as

$$T_k^{PPM} = \widetilde{O}\left(\varepsilon_{k+1}^{-2}\right)$$

We now define the *first* power ALM (outer) iteration $K_A$ for which

$$\varepsilon_A \geq \varepsilon_{A,K_A+1} = \left( \frac{\nabla f_{\max} + J_{A\max} y_{\max} + \varepsilon_1}{v \beta_{K_A}} \right)^{\frac{1}{\nu}} := \frac{Q_A}{\beta_{K_A}^{1/\nu}} \tag{52}$$

and from (29) we have that

$$K_A = \left\lceil \log_\omega \left( \frac{Q'_A}{\varepsilon_A^\nu} \right) \right\rceil + 2.$$

Hence, we require at most $T_A$ total FGM iterations to obtain a point $x^{K_A+1}$, where

$$
\begin{aligned}
T_A &= \sum_{k=1}^{K_A} T_k^{PPM} T_k^{FGM} \\
&= \sum_{k=1}^{K_A} \widetilde{O}\left( \varepsilon_{k+1}^{-2} \beta_k^{\frac{1-\nu}{1+3\nu}\frac{1+\nu}{\nu} + \frac{2}{1+3\nu}} \varepsilon_{k+1}^{-\frac{1+\nu}{\nu}\frac{1-\nu}{1+3\nu}} \right) = \sum_{k=1}^{K_A} \widetilde{O}\left( \beta_k^{\frac{1-\nu}{1+3\nu}\frac{1+\nu}{\nu} + \frac{2}{1+3\nu}} \beta_k^{2 + \frac{1+\nu}{\nu}\frac{1-\nu}{1+3\nu}} \right) \\
&= \sum_{k=1}^{K_A} \widetilde{O}\left( \beta_k^{2 + 2\frac{1-\nu}{1+3\nu}\frac{1+\nu}{\nu} + \frac{2}{1+3\nu}} \right) \\
&\leq \widetilde{O}\left( K_A \beta_{K_A}^{2 + 2\frac{1-\nu}{1+3\nu}\frac{1+\nu}{\nu} + \frac{2}{1+3\nu}} \right) = \widetilde{O}\left( \beta_{K_A}^{2 + 2\frac{1-\nu}{1+3\nu}\frac{1+\nu}{\nu} + \frac{2}{1+3\nu}} \right)
\end{aligned}
$$

By substitution of $\beta_{K_A} = \omega \beta_{K_A-1} < \frac{Q_A}{\varepsilon_A^\nu}$ this yields

$$T_A = \widetilde{O}\left( \varepsilon_A^{-2\nu - 2\frac{1-\nu}{1+3\nu}(1+\nu) - \frac{2\nu}{1+3\nu}} \right).$$

In a similar way to the proof of Theorem 2 we can define the *first* power ALM (outer) iteration $K_\varphi$ for which

$$\varepsilon_\varphi \geq \varepsilon_{\varphi,K_\varphi+1} = \frac{1}{\beta_{K_\varphi}} \left( \lambda + J_{A\max} \sigma_1 \frac{\nabla f_{\max} + J_{A\max} y_{\max} + \varepsilon_1}{v} \right) := \frac{Q_f}{\beta_{K_\varphi}}$$

and obtain

$$K_\varphi = \left\lceil \log_\omega \left( \frac{Q'_f}{\varepsilon_\varphi} \right) \right\rceil + 2.$$

$$T_\varphi = \widetilde{O}\left( \beta_{K_\varphi}^{2 + 2\frac{1-\nu}{1+3\nu}\frac{1+\nu}{\nu} + \frac{2}{1+3\nu}} \right)$$

By substitution of $\beta_{K_\varphi} = \omega \beta_{K_\varphi-1} < \frac{Q_f}{\varepsilon_\varphi}$ this yields

$$T_\varphi = \widetilde{O}\left( \varepsilon_\varphi^{-2 - 2\frac{1-\nu}{1+3\nu}\frac{1+\nu}{\nu} - \frac{2}{1+3\nu}} \right).$$

$\square$

# B  Additional experiments

## B.1  Generalized eigenvalue problem

We consider the generalized eigenvalue problem (GEVP)

$$\min_{x\in\mathbb{R}^n} x^\top C x \quad \text{s.t.} \quad x^\top B x = 1,$$

where $B, C \in \mathbb{R}^{n\times n}$ are symmetric matrices and $B$ is positive definite. Clearly, the GEVP is of the form (P), and satisfies the regularity condition Assumption 4 Sahin et al. (2019). We sample the entries of a matrix $\widehat{C} \in \mathbb{R}^{n\times n}$ from a Gaussian $\mathcal{N}(0, 0.1)$ and define $C := \frac{1}{2}(\widehat{C} + \widehat{C}^\top)$. The matrix $B$ is defined as $Q^\top Q$, where $Q$ is the orthonormal matrix in the QR-decomposition of a random matrix with entries sampled uniformly from the unit interval. We use the classical accelerated proximal gradient method (APGM) by Beck & Teboulle (2009) as an inner solver and tune its step size to $0.5/(10\|C\| + 5000 + 500\beta)$. Although APGM has no convergence guarantees that fully cover our setting, i.e., nonconvex and Hölder-smooth objectives, it appears that convergence issues can be mitigated by sufficiently decreasing the step size. We follow the tuning of Li et al. (2021) $\beta_1 = 0.01, \omega = 3, \lambda = 1, \sigma_1 = 10$, and impose a maximum of $N = 10^5$ APGM iterations per subproblem.

Table 4 reports the number of gradient calls that power ALM requires to attain an $(\varepsilon_\varphi, \varepsilon_A)$-stationary point with $\varepsilon_\varphi = \varepsilon_A = 10^{-3}$ for various powers $\nu \in (0, 1]$. Also the constraint violation $\|A(x)\|$ and the suboptimality $|f(x) - f^*|$ are listed. Every trial denotes a random realization of the GEVP with $n = 500$, and the first trial is further illustrated in Figure 4. We observe that smaller values of $\nu$ perform significantly better than larger values, with $\nu = 0.4$ requiring an order of magnitude fewer gradient evaluations than $\nu = 1$. The figure corresponding to the first realization confirms that both constraint violation and suboptimality decrease steadily, even for a small power $\nu = 0.4$.

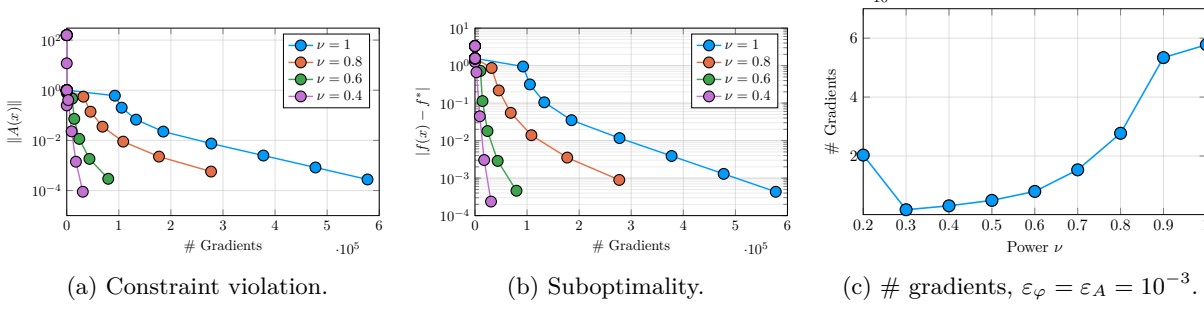

(a) Constraint violation.        (b) Suboptimality.        (c) # gradients, $\varepsilon_\varphi = \varepsilon_A = 10^{-3}$.

Figure 4: Comparison of the proposed power ALM with various powers $\nu$ on solving a representative GEVP with $n = 500$. The case $\nu = 1$ corresponds to the iALM from Sahin et al. (2019).

Table 4: Performance of power ALM with APGM inner solver on solving random GEVPs of size $n = 500$.

| | $\nu = 0.2$ | | | $\nu = 0.4$ | | | $\nu = 0.6$ | | | $\nu = 0.8$ | | | $\nu = 1.0$ | | |
|---|---|---|---|---|---|---|---|---|---|---|---|---|---|---|---|
| trial | const. viol. | subopt | # grads | const. viol. | subopt. | # grads | const. viol. | subopt | # grads | const. viol. | subopt | # grads | const. viol. | subopt | # grads |
| 1 | $6.1\cdot10^{-4}$ | $9.6\cdot10^{-4}$ | $2\cdot10^5$ | $9\cdot10^{-5}$ | $2.4\cdot10^{-4}$ | $30,808$ | $2.9\cdot10^{-4}$ | $4.6\cdot10^{-4}$ | $79,695$ | $5.7\cdot10^{-4}$ | $8.9\cdot10^{-4}$ | $2.8\cdot10^5$ | $2.8\cdot10^{-4}$ | $4.3\cdot10^{-4}$ | $5.8\cdot10^5$ |
| 2 | $1.6\cdot10^{-4}$ | $2.5\cdot10^{-4}$ | $1\cdot10^5$ | $1\cdot10^{-4}$ | $4\cdot10^{-4}$ | $31,540$ | $3.1\cdot10^{-4}$ | $4.9\cdot10^{-4}$ | $1\cdot10^5$ | $6.2\cdot10^{-4}$ | $9.6\cdot10^{-4}$ | $2.8\cdot10^5$ | $3\cdot10^{-4}$ | $4.6\cdot10^{-4}$ | $6.1\cdot10^5$ |
| 3 | $1.6\cdot10^{-4}$ | $2.6\cdot10^{-4}$ | $1\cdot10^5$ | $1\cdot10^{-4}$ | $2.3\cdot10^{-4}$ | $31,498$ | $3.2\cdot10^{-4}$ | $5.1\cdot10^{-4}$ | $1.1\cdot10^5$ | $1.6\cdot10^{-4}$ | $2.6\cdot10^{-4}$ | $3.9\cdot10^5$ | $3.1\cdot10^{-4}$ | $4.9\cdot10^{-4}$ | $6.5\cdot10^5$ |
| 4 | $6.1\cdot10^{-4}$ | $9.8\cdot10^{-4}$ | $2\cdot10^5$ | $9.1\cdot10^{-5}$ | $2.5\cdot10^{-4}$ | $31,926$ | $3\cdot10^{-4}$ | $4.7\cdot10^{-4}$ | $82,429$ | $5.8\cdot10^{-4}$ | $9.3\cdot10^{-4}$ | $2.8\cdot10^5$ | $2.8\cdot10^{-4}$ | $4.5\cdot10^{-4}$ | $6.4\cdot10^5$ |
| 5 | $1.7\cdot10^{-4}$ | $2.6\cdot10^{-4}$ | $1\cdot10^5$ | $1\cdot10^{-4}$ | $3.2\cdot10^{-4}$ | $28,301$ | $3.1\cdot10^{-4}$ | $4.9\cdot10^{-4}$ | $88,925$ | $6.2\cdot10^{-4}$ | $9.6\cdot10^{-4}$ | $2.8\cdot10^5$ | $3\cdot10^{-4}$ | $4.6\cdot10^{-4}$ | $6.2\cdot10^5$ |
| 6 | $1.9\cdot10^{-4}$ | $2.9\cdot10^{-4}$ | $1\cdot10^5$ | $1.1\cdot10^{-4}$ | $7\cdot10^{-4}$ | $36,529$ | $3.2\cdot10^{-4}$ | $5.1\cdot10^{-4}$ | $1.6\cdot10^5$ | $1.7\cdot10^{-4}$ | $2.6\cdot10^{-4}$ | $3.9\cdot10^5$ | $3.2\cdot10^{-4}$ | $4.9\cdot10^{-4}$ | $6.6\cdot10^5$ |
| 7 | $1.8\cdot10^{-4}$ | $2.8\cdot10^{-4}$ | $1\cdot10^5$ | $4.7\cdot10^{-5}$ | $8.6\cdot10^{-5}$ | $1.5\cdot10^5$ | $3.2\cdot10^{-4}$ | $5\cdot10^{-4}$ | $2.1\cdot10^5$ | $6.3\cdot10^{-4}$ | $9.8\cdot10^{-4}$ | $3.8\cdot10^5$ | $3\cdot10^{-4}$ | $4.7\cdot10^{-4}$ | $6.6\cdot10^5$ |
| 8 | $1.8\cdot10^{-4}$ | $2.8\cdot10^{-4}$ | $1\cdot10^5$ | $1\cdot10^{-4}$ | $2.8\cdot10^{-4}$ | $33,362$ | $3.1\cdot10^{-4}$ | $4.8\cdot10^{-4}$ | $1\cdot10^5$ | $6.2\cdot10^{-4}$ | $9.5\cdot10^{-4}$ | $3\cdot10^5$ | $3\cdot10^{-4}$ | $4.6\cdot10^{-4}$ | $6.5\cdot10^5$ |
| 9 | $1.6\cdot10^{-4}$ | $2.4\cdot10^{-4}$ | $1\cdot10^5$ | $4.7\cdot10^{-5}$ | $6.7\cdot10^{-4}$ | $1.3\cdot10^5$ | $3.1\cdot10^{-4}$ | $5.7\cdot10^{-4}$ | $2.3\cdot10^5$ | $6.1\cdot10^{-4}$ | $9.5\cdot10^{-4}$ | $4.4\cdot10^5$ | $2.9\cdot10^{-4}$ | $4.6\cdot10^{-4}$ | $7.6\cdot10^5$ |
| 10 | $6.2\cdot10^{-4}$ | $9.6\cdot10^{-4}$ | $2\cdot10^5$ | $1\cdot10^{-4}$ | $2.2\cdot10^{-4}$ | $25,497$ | $3\cdot10^{-4}$ | $4.7\cdot10^{-4}$ | $86,236$ | $5.9\cdot10^{-4}$ | $9.3\cdot10^{-4}$ | $2.7\cdot10^5$ | $2.9\cdot10^{-4}$ | $4.5\cdot10^{-4}$ | $6.2\cdot10^5$ |
| avg. | $3\cdot10^{-4}$ | $4.8\cdot10^{-4}$ | $1.3\cdot10^5$ | $\mathbf{8.9\cdot10^{-5}}$ | $\mathbf{3.4\cdot10^{-4}}$ | $\mathbf{52{,}839.4}$ | $3.1\cdot10^{-4}$ | $5\cdot10^{-4}$ | $1.2\cdot10^5$ | $5.2\cdot10^{-4}$ | $8.1\cdot10^{-4}$ | $3.3\cdot10^5$ | $3\cdot10^{-4}$ | $4.6\cdot10^{-4}$ | $6.4\cdot10^5$ |

## C  Inner solvers

For completeness, this section summarizes the *universal problem-parameter free accelerated gradient* method (Ghadimi et al., 2019, Algorithm 1) method and the *fast gradient method* (Devolder et al., 2014, Algorithm 3), which are, respectively, used as inner solvers in sections 2 and 3.

First, we consider the ALM subproblem in Algorithm 1 Line 2, which are of the form

$$\underset{x \in \mathbb{R}^n}{\text{minimize}} \, \Psi(x) \equiv \psi(x) + g(x). \tag{53}$$

Here, $\psi : \mathbb{R}^n \to \mathbb{R}$ is Hölder-smooth and $g : \mathbb{R}^n \to \overline{\mathbb{R}}$ is convex. Algorithm 3 specifies the UPFAG method (Ghadimi et al., 2019, Algorithm 1) for problems of the form (53).

---

**Algorithm 3** Unified problem-parameter free accelerated gradient (UPFAG) method (Ghadimi et al., 2019)

**Require:** $x_0 \in \mathbb{R}^n$, $\gamma_1, \gamma_2, \gamma_3 \in (0, 1)$, accuracy parameter $\delta > 0$.

1: Set $x_0^{\text{ag}} = x_0$, and $\Lambda_0 = 0$.
2: **for** $k = 1, 2, \ldots$ **do**
3:     Choose initial stepsize $\hat{\lambda}_k > 0$ and find the smallest integer $\tau_{1,k} \geq 0$ such that with

$$\eta_k = \hat{\lambda}_k \gamma_1^{\tau_{1,k}}, \quad \lambda_k = \frac{\eta_k + \sqrt{\eta_k^2 + 4\eta_k \Lambda_{k-1}}}{2}, \quad \alpha_k = \frac{\lambda_k}{\Lambda_k} \quad \text{and} \quad \Lambda_k = \sum_{i=1}^{k} \lambda_i,$$

the solutions obtained by

$$x_k^{\text{md}} = (1 - \alpha_k)x_{k-1}^{\text{ag}} + \alpha_k x_{k-1},$$
$$x_k = \mathbf{prox}_{\lambda_k g}(x_{k-1} - \lambda_k \nabla \psi(x_k^{\text{md}}))$$
$$\tilde{x}_k^{\text{ag}} = (1 - \alpha_k)x_{k-1}^{\text{ag}} + \alpha_k x_k,$$

satisfy the condition

$$\psi(\tilde{x}_k^{\text{ag}}) \leq \psi(x_k^{\text{md}}) + \alpha_k \langle \nabla \psi(x_k^{\text{md}}), x_k - x_{k-1} \rangle + \frac{\alpha_k}{2\lambda_k} \|x_k - x_{k-1}\|^2 + \delta \alpha_k.$$

4:     Choose initial stepsize $\hat{\beta}_k > 0$ and find the smallest integer $\tau_{2,k} \geq 0$ such that with

$$\beta_k = \hat{\beta}_k \gamma_2^{\tau_{2,k}} \quad \text{and} \quad \bar{x}^{\text{ag}} = \mathbf{prox}_{\beta_k g}(x_{k-1}^{\text{ag}} - \beta_k \nabla \psi(x_{k-1}^{\text{ag}}))$$

satisfy the condition

$$\Psi(\bar{x}_k^{\text{ag}}) \leq \Psi(x_{k-1}^{\text{ag}}) - \frac{\gamma_3}{2\beta_k} \|\bar{x}_k^{\text{ag}} - x_{k-1}^{\text{ag}}\|^2 + \frac{1}{k}.$$

5:     Choose $x_k^{\text{ag}}$ such that

$$\Psi(x_k^{\text{ag}}) = \min \left\{ \Psi(x_{k-1}^{\text{ag}}), \Psi(\bar{x}_k^{\text{ag}}), \Psi(\tilde{x}_k^{\text{ag}}) \right\}.$$

6: **end for**

---

Second, we consider the proximal point updates in Algorithm 2 Line 2, which are of the form

$$\underset{x \in \mathcal{X}}{\text{minimize}} \, F(x). \tag{14 rev.}$$

Here, $F : \mathbb{R}^n \to \mathbb{R}$ is $(H_f, \nu)$-Hölder-smooth and $\rho$-strongly convex with $H_f \geq 0, \rho > 0$ and $\nu \in (0, 1]$. The set $\mathcal{X} \subseteq \mathbb{R}^n$ is closed and convex. Algorithm 4 specializes (Devolder et al., 2014, Algorithm 3) to problems of the form (14) for the standard Euclidean *prox-function*.

---

**Algorithm 4** Fast gradient method for strongly convex, Hölder smooth functions (Devolder et al., 2014)

---

**Require:** $x_0 \in \mathcal{X}$, accuracy parameter $\delta > 0$, constants $\rho > 0, H_f \geq 0, \nu \in (0, 1]$.

1: Set $\mu = \rho$ and $L = H_f \left( \frac{H_f}{2\delta} \frac{1-\nu}{1+\nu} \right)^{\frac{1-\nu}{1+\nu}}$, and let $\{\alpha_k\}_{k \geq 0}$ be such that $\alpha_0 = 1$ and $L + \mu \sum_{i=0}^{k} \alpha_i = \frac{L\alpha_{k+1}^2}{\sum_{i=0}^{k+1} \alpha_i}$.

2: **for** $k = 0, 1, \ldots$ **do**

3:   $y_k = \arg\min_{x \in \mathcal{X}} \left\{ \langle \nabla F(x_k), x - x_k \rangle + \frac{L}{2} \|x - x_k\|^2 \right\}$.

4:   $z_k = \arg\min_{x \in \mathcal{X}} \left\{ \frac{L}{2} \|x - x_0\|^2 + \sum_{k=0}^{k} \alpha_i \left[ \langle \nabla F(x_i), x - x_i \rangle + \frac{\mu}{2} \|x - x_i\|^2 \right] \right\}$.

5:   $x_{k+1} = \tau_k z_k + (1 - \tau_k) y_k$ where $\tau_k = \frac{\alpha_{k+1}}{\sum_{i=0}^{k+1} \alpha_i}$.

6: **end for**

---

Note that the first step is a standard projected gradient update $y_k = \mathbf{proj}_{\mathcal{X}}(x_k - \frac{1}{L}\nabla F(x_k))$, whereas the second step can be written as

$$z_k = \mathbf{proj}_{\mathcal{X}} \left( \frac{Lx_0 + \mu \sum_{i=0}^{k} \alpha_i x_i - \sum_{i=0}^{k} \alpha_i \nabla F(x_i)}{L + \mu \sum_{i=0}^{k} \alpha_i} \right).$$

We highlight that per iteration Algorithm 4 only requires a single gradient evaluation of $F$ (at $x_k$), and two projections onto $\mathcal{X}$.

