# OpenReview forum: "The inexact power augmented Lagrangian method for constrained nonconvex optimization"
_TMLR — Accepted by TMLR_

### Review · Reviewer_uuHq · 2025-08-01

**Summary Of Contributions:**

### Strengths

1. The paper introduces an unconventional inexact augmented Lagrangian method (iALM), using a Euclidean norm raised to a power between 1 and 2. The paper claims that the choice can provide more flexibility and improved performance in balancing primal and dual complexities.
2. Under mild regularity assumptions, the authors establish rigorous convergence results to first-order stationary points, which can be applied to quantify the trade-off between primal feasibility and dual optimality.
3. An inexact proximal point method tailored for weakly-convex and Hölder-smooth subproblems is presented with convergence guarantees.
4. The paper is well-organized and well-written and the convergence analysis is overall correct, to the best of my knowledge.
5. Numerical experiments are provided to validate the theoretical findings.

### Weaknesses
1. *Motivation for the unconventional iALM approach is not fully convincing*. The contribution appears incremental as the convergence analysis heavily leverages existing results, and numerical experiments exhibit only marginal improvement.
2. *Excessive assumptions with unclear relation to existing literature*. The paper introduces numerous assumptions without adequately clarifying their necessity or explicitly contrasting them with standard assumptions in prior works.
3. *Unclear rationale behind defining the $(\epsilon_\phi, \epsilon_A)$-stationary point*. The introduction of the joint stationary point criterion seems arbitrary or insufficiently motivated from a theoretical or practical perspective.
4. *Gap between theoretical analysis and numerical results*. APGM, used as an inner solver in numerical experiments, may not converge under the stated theoretical conditions, causing discrepancies between theoretical guarantees and experimental validations.

**Additional Comments:**

NA

**Audience:**

Yes

**Audience Explanation:**

The (iALM) is among the most influential and widely-used algorithms for solving large-scale convex and nonconvex nonlinear programming problems, particularly due to its strong theoretical foundations and broad applicability. It has found extensive use in diverse fields including optimization, operations research, and modern machine learning applications. Given its practical importance, analyzing the convergence properties and complexity of iALM, especially under realistic and relaxed assumptions, remains a significant and active area of research. Establishing rigorous complexity guarantees and convergence rates provides deeper theoretical insights and drives the design of efficient computational algorithms, contributing directly to solving increasingly challenging real-world optimization problems.

**Claims And Evidence:**

No

**Claims Explanation:**

Some claims are not fully supported by clear evidence. Please see the weaknesses above.

**Requested Changes:**

1. Clarify specific scenarios or classes of problems where the proposed unconventional augmenting terms clearly outperform existing methods. Provide more compelling numerical evidence showcasing significant advantages in constraint satisfaction or complexity reduction, highlighting real-world relevance.
2. Include a dedicated discussion (e.g., a table summarizing the differences) explicitly comparing each assumption to those commonly adopted in related literature. Justify explicitly why each newly introduced or relaxed assumption is necessary and how it contributes uniquely to the theoretical or practical improvement of the method.
3. Provide a clear motivation or practical scenario demonstrating why considering separate primal and dual tolerances, e.g., when it is reduced to the classical $\epsilon$-stationary point, what are the main differences.
4. Clearly state and justify the choice of APGM, such as conditions under which convergence is assured or acknowledging the potential limitations explicitly. Alternatively, conduct additional experiments using theoretically guaranteed solvers (e.g., the methods analyzed in the paper) to close the gap and reinforce theoretical claims.

---

> ### Author Response · Authors · 2025-08-28
>
> We thank the reviewer for their feedback. Below, we address the stated weaknesses and requested changes.
>
> **W1/Q1:** We respectfully disagree with the claim that the experiments only exhibit marginal improvement. Figures 1c and 2c show that for clustering problems a good power $\nu$ may half the number of UPFAG iterations. Likewise, we observe significant speedups for both quadratic programming and generalized eigenvalue problems.
>
> Nevertheless, we would like to stress that our contribution is primarily theoretical of nature and focuses on the complexity analysis of iALMs: (i) we show that sharper penalty terms $\nu < 1$ are actually reflected in the complexity (and thus yield faster constraint satisfaction); and (ii) we present novel proof techniques to relax various assumptions compared to existing works, even for $\nu = 1$ (see also Q2).
>
> In section 1.1 we motivated penalty terms with $\nu < 1$, at least in the convex case, through their *equivalence* to *high-order proximal point methods*, which are the subject of active research.
> To further motivate such penalty terms, we added an explicit connection to *sharp Lagrangians*, which notably support exact penalty representations.
> To our knowledge, it is unique that this is reflected in a complexity analysis.
>
> We believe that our theoretical contributions are far from incremental. For example: (i) Lemma 2 and its proof are novel even for $\nu = 1$ in *establishing boundedness of the iterates*, whereas related works simply assume bounded domain; (ii) we handle *generic convex terms $g$* under an appropriate regularity condition (Assumption 4) by exploiting the structure of $\partial g$ (cf. proof of Theorem 1); and (iii) we design a proximal point method which exploits qualitatively different lower and upper bounds -- from weak convexity and Hölder smoothness respectively.
>
> **W2/Q2:** We included an additional paragraph and a table at the end of section 1 to summarize in detail how our assumptions relate to, and in fact *relax*, those in other works. Unlike existing works, we do not assume bounded domain or bounded iterates, and we relax Lipschitz smoothness assumptions to (truly local) Hölder smoothness assumptions. Moreover, we handle generic convex terms $g$, rather than just indicators of convex sets. The few existing works which handle generic convex terms either require additional conditions that may be hard to verify, or use a regularity condition which has been questioned in the literature.
>
> **W3/Q3:** Analyses of existing iALMs (with $\nu = 1$) consider $\varepsilon_\varphi = \varepsilon_A$, because only $\min \{\varepsilon_\varphi, \varepsilon_A\}$ affects the obtained complexity. Yet, for $\nu < 1$ primal and dual tolerances have a distinct effect on the overall complexity, and our work appears unique in describing this. We highlight that the limiting case $\nu = 0$ recovers a sharp Lagrangian which supports an exact penalty representation. Thus as $\nu$ is decreased from $1$ to $0$, one expects to obtain faster constraint satisfaction, which is precisely what we obtain.
>
> Also in practice distinct tolerances are relevant, for example when the objective and the constraints are scaled very differently. In fact, general-purpose solvers typically support distinct primal and dual tolerances for this reason, see for example the references below for IPOPT, MOSEK, and GUROBI.
>
> **W4/Q4:** We have updated the experiments in the main text to use the UPFAG method as inner solver, i.e., the setting covered by Theorem 2. In this way, there are no more discrepancies between the theoretical results and the experimental validations. Qualitatively, the experimental results remain similar.
>
> **References**
>
> IPOPT: https://coin-or.github.io/Ipopt/OPTIONS.html
>
> MOSEK: https://docs.mosek.com/11.0/cxxfusion/parameters.html
>
> GUROBI: https://docs.gurobi.com/projects/optimizer/en/current/concepts/numericguide/tolerances_scaling.html

---

> ### Comment · Reviewer_uuHq · 2025-09-09
>
> The authors have addressed my previous concerns. I have no further comment.

---

### Review · Reviewer_ewE5 · 2025-08-05

**Summary Of Contributions:**

This paper introduces an inexact power augmented Lagrangian method (iALM) for solving constrained nonconvex optimization problems with nonlinear equality constraints $ A(x) = 0 $. The method extends the classical augmented Lagrangian by introducing a penalty term of the form $ ||A(x)||^{1+\nu} $ with $ \nu \in (0, 1] $. When $ \nu = 1 $, the approach naturally reduces to the classical augmented Lagrangian method. This generalization offers a novel trade-off between primal and dual residual tolerances, providing additional flexibility in balancing constraint satisfaction and suboptimality reduction.

The proposed iALM achieves linear convergence in the number of outer iterations to a stationary point, while the constraint violation decreases at a faster rate when $ \nu < 1 $.  For problems where $ \nabla f $ and $ A $ are locally Hölder continuous, the subproblems are solved using the existing UPFAG algorithm. This recovers the state-of-the-art complexity result of $ \tilde{\mathcal{O}}(\epsilon^{-4}) $ for the overall method.

Additionally, under a stronger constraint qualification, the paper introduces a new subroutine for solving the Hölder-smooth subproblems. This subroutine combines the proximal point algorithm with an accelerated gradient method, leading to improved overall complexity results. Specifically, for $ \nu = 1 $, the method achieves $ \tilde{\mathcal{O}}(\epsilon^{-3}) $ complexity, matching the best-known rates in the literature. For linear constraints, the complexity is further improved to $ \tilde{\mathcal{O}}(\epsilon^{-2.5}) $.

**Audience:**

Yes

**Audience Explanation:**

The findings of this paper would likely interest TMLR's audience, as it introduces an inexact augmented Lagrangian method with theoretical advancements and practical implications for constrained nonconvex optimization—a topic relevant to researchers in machine learning and optimization.

**Claims And Evidence:**

Yes

**Claims Explanation:**

Most claims in the paper are supported by rigorous proofs. However, I have the following questions and concerns:

1. In Lemma 1, the boundedness of the dual variable $ y_k $ is established by using a very small, normalized step size. However, I find it unusual that, having a bound for $ y_k $, then the Lagrangian term $ \langle A(x), y \rangle $ appears to play a minimal role in the proof of Theorem 1. It seems to me that the main force enforcing the constraint comes from the penalty term $ \|A(x)\|^{\nu+1} $. This raises the question: what happens if we simply fix $ y_k = y $ for all iterations? Would the convergence results in Theorem 1 still hold? Additionally, in Theorem 1, the algorithm is shown to converge at a rate of $ \omega^{-k} $, where $ \omega $ can be arbitrarily large. Does this imply that the algorithm could theoretically converge in a single iteration if $ \beta $ is chosen to be sufficiently large?

2. Assumption 1 only requires local Hölder continuity/smoothness, meaning that the constants $ H_f $ and $ H_A $ depend on the local compact set $ S $. For the inner complexity results (Lemma 3 and Lemma 4), I am unclear about the set $ S $. Specifically, with respect to which set $ S $ are the constants $ H_f $ and $ H_A $ defined?

3. What are the implications when $A$ is linear and Assumption 5 holds?

**Requested Changes:**

Please refer to my questions above. My main concern is the effect of updating y after simply bounding it.

---

> ### Author Response · Authors · 2025-08-28
>
> We thank the reviewer for their feedback.
>
> **Q1:** You are right in your observation that, at least in the complexity analysis, the penalty term is the main force in ensuring feasibility. Theorem 1 still holds when we fix the multipliers. In fact, our analysis can easily be adapted to *penalty* methods, i.e., Algorithm 1 with the term $\langle A(x), y\rangle$ omitted from the objective of the inner problems. Similar observations hold for the iALMs by Sahin et al. (2019) and by Li et al. (2021) and are explicitly mentioned by their respective authors.
> Given that augmented Lagrangian methods are well-known to outperform penalty methods in practice, it is rather remarkable that the best-known complexities of iALMs match those of the corresponding penalty methods. This also holds for our results.
>
> Regarding $\omega$ which can be chosen arbitrarily large, this indeed implies Algorithm 1 could be made to converge in a single *outer* iteration. The catch here is that a large $\omega$ results in a large smoothness constant of the augmented Lagrangian (cf. Lemma 3), which in turn requires a large number of *inner* iterations. This is similar to how an inexact proximal point method with sufficiently small step size can be made to converge in a single iteration. The catch is that the inner problem becomes as hard as the original problem.
>
> **Q2:** Smoothness constants like $H_f$ and $H_A$ are assumed relative to any compact set $\mathcal{S}$. Lemma 3 then provides an expression for $H_\beta$, the smoothness constant of the augmented Lagrangian, relative to any compact set $\mathcal S$ in terms of the smoothness constants $H_f$ and $H_A$ corresponding to that compact set $\mathcal S$.
>
> Lemma 2 establishes that the iterates of Algorithm 1 remain in the compact initial sublevel set $\mathcal L_{\beta_1,y^1}(x^1)$ of the augmented Lagrangian. Therefore, in Lemma 4 and Theorem 2 the smoothness constants are defined relative to the compact set $\mathcal S = \mathcal L_{\beta_1,y^1}(x^1)$. We add a remark to clarify this.
>
> **Q3:** If $A(\cdot)$ is a linear map, then the aspects of Assumption 5 that relate to $A(\cdot)$, like Lipschitz-smoothness of $A(\cdot)$, are automatically satisfied. Only the assumptions on the objective terms $f$ and $g$ remain to be verified.
> For linear maps $A(\cdot)$, it follows under Assumption 4 (the regularity condition) and Assumption 5 that an improved complexity result is attained, as described in Theorem 6.

---

> > ### Comment · Reviewer_ewE5 · 2025-09-10
> >
> > Thank you for the authors’ reply. In Assumption 1, do the values of H_f and H_A depend on S?

---

> > > ### Author Response · Authors · 2025-09-14
> > >
> > > Indeed, in Assumption 1 the constants $H_f$ and $H_A$ depend on the given set $S$.
> > >
> > > Lemma 2 shows that the iterates of Algorithm 1 remain in a compact initial sublevel set $\mathcal{L}\_{{\beta_1, y^1}}$; hence the constants $H_f$ and $H_A$ in Lemma 4 and Theorem 2 are defined with respect to the compact sublevel set $\mathcal{L}_{\beta_1, y^1}$. This improves upon existing works, which assume compact domain $\mathcal X$ and define $H_f$ and $H_A$ with respect to this (larger) set $\mathcal X$.

---

> > > > ### Comment · Reviewer_ewE5 · 2025-09-15
> > > >
> > > > Thank you for the reply. Please state it explicitly in Lemma 3.
> > > >
> > > > I do not have further questions.

---

> > > > > ### Author Response · Authors · 2025-09-15
> > > > >
> > > > > Thank you for the suggestion; we have incorporated it.

---

### Review · Reviewer_hcPb · 2025-08-21

**Summary Of Contributions:**

This paper considers nonconvex optimization with nonlinear constraints and a potentially sharper penalty function. The authors first propose a novel inexact augmented Lagrangian method for such problem, followed by a novel inexact proximal point method for solving the inner problems. The paper provides a detailed complexity analysis for the proposed methods and validates the empirical performance of unconventional augmenting terms.

**Audience:**

Yes

**Audience Explanation:**

Nonconvex minimization problem with nonlinear constraints is an important problem in the fields of optimization and machine learning. Researchers in optimization area, in particular, will likely be interested in the findings of this paper.

**Broader Impact Concerns:**

No broader impact concerns

**Claims And Evidence:**

Yes

**Claims Explanation:**

I do not check the proofs in the appendix, but the theoretical results seem reasonable and solid. The complexity of Algorithm 1 matches the one from Sahin et al. (2019) in the Lipschitz smooth setting. If the inner subproblem is solve by Algorithm 2, the authors show that the total complexity nearly matches the complexity result in (Li et al., 2021).

**Requested Changes:**

1. To enhance the self-containment of the paper, I recommend that the authors include the UPFAG and FGM algorithms in the appendix.
2. Follow the previous point, I think the paper should provide the complexity of first-order oracle instead of the complexity of the iterations of the UPFAG and FGM algorithms. Readers may not be aware of how many first-order oracle calls occur in a single iteration of these algorithms.
3. It seems that the paper does not provide a definition for lsc functions. Some readers may not be familiar with this concept, so it would be beneficial to include a brief explanation.
4. The paper may discuss whether Algorithm 1 can finds a second-order stationary point if a second-order solver is used for the inner iterates.

---

> ### Author Response · Authors · 2025-08-28
>
> We thank the reviewer for their feedback.
>
> **Q1:** Thank you for this suggestion. We have included a novel section in the appendix (C) which summarizes the UPFAG and FGM methods in the context of this work.
>
> **Q2:** This is a good point. We have added remarks after Theorems 2 and 5, respectively, to clarify that these theorems also describe the complexity in terms of first-order oracle calls. In particular, the average number of first-order oracles calls per UPFAG iteration can be bounded by a constant as described by Ghadimi et al. (2019, End of Section 2). FGM requires one gradient and two projections per iteration.
>
> **Q3:** We included a brief definition and a reference for the interested readers.
>
> **Q4:** This is an interesting question. First, a *second-order* stationary point of the problem (P), with a *nonsmooth* composite objective, must be properly defined, for example based on the necessary conditions of optimality in (Rockafellar & Wets, 1998, Theorem 13.24). For the case $g \equiv 0$, Sahin et al. (2021) provide a complexity analysis in which they consider the smallest eigenvalue of $\nabla_{xx}^2 L_\beta(x, y)$ as a second-order stationarity measure. This is the natural second-order stationarity measure of the primal ALM subproblems, but it remains unclear how this translates to (approximate) stationarity of the original problem (P). In fact, even verifying approximate second-order stationarity in the presence of *linear* constraints is NP-hard (Nouiehed et al., 2018).
> More specific to our setup, we would also require a complexity analysis of a second-order solver under *Hölder*-smoothness, instead of the usual Lipschitz-smoothness.
> Unfortunately, we are not aware of such results.
> Overall, we believe this direction requires significant work that goes beyond the scope of our current submission.
>
> **Additional References**
>
> Nouiehed, Maher, Jason D. Lee, and Meisam Razaviyayn. "Convergence to second-order stationarity for constrained non-convex optimization." arXiv preprint arXiv:1810.02024 (2018).

---

### Decision · Action_Editor_cX2M · 2025-10-07

**Recommendation:** Accept as is

**Audience:**

Yes

**Audience Explanation:**

The studied method is one of the go-to choices of large-scale convex and nonconvex nonlinear programming problems, so it would naturally interest a significant part of TMLR audience.

**Claims And Evidence:**

Yes

**Claims Explanation:**

The reviewers unanimously agreed the proofs are supported by rigorous proofs. The authors also supported the theory with a proper numerical evaluation.